# Ecological design of augmentation improves helicopter ship landing maneuvers: An approach in augmented virtuality

**Antoine H. P. Morice**[1]*, **Thomas Rakotomamonjy**[2], **Julien R. Serres**[1], **Franck Ruffier**[1]

**1** Aix Marseille Univ, CNRS, ISM, Marseille, France, **2** DTIS, ONERA, Salon Cedex Air, France

\* antoine.morice@univ-amu.fr

**Data Availability Statement:** All original data files are available from the repository: HelicopterShipLanding-TauDot_data at https://

## Abstract

Helicopter landing on a ship is a visually regulated "rendezvous" task during which pilots must use fine control to land a powerful rotorcraft on the deck of a moving ship tossed by the sea while minimizing the energy at impact. Although augmented reality assistance can be hypothesized to improve pilots' performance and the safety of landing maneuvers by guiding action toward optimal behavior in complex and stressful situations, the question of the optimal information to be displayed to feed the pilots' natural information-movement coupling remains to be investigated. Novice participants were instructed to land a simplified helicopter on a ship in a virtual reality simulator while minimizing energy at impact and landing duration. The wave amplitude and related ship heave were manipulated. We compared the benefits of two types of visual augmentation whose design was based on either solving cockpit-induced visual occlusion problems or strengthening the online regulation of the deceleration by keeping the current $\dot{\tau}$ variable around an ideal value of -0.5 to conduct smooth and efficient landing. Our results showed that the second augmentation, ecologically grounded, offers benefits at several levels of analysis. It decreases the landing duration, improves the control of the helicopter displacement, and sharpens the sensitivity to changes in $\dot{\tau}$. This underlines the importance for designers of augmented reality systems to collaborate with psychologists to identify the relevant perceptual-motor strategy that must be encouraged before designing an augmentation that will enhance it.

## 1. Introduction

Landing a helicopter on a ship's deck is a highly complex and demanding task. A first difficulty is linked to the number of parameters and degrees of freedom of movement to monitor (6 for the helicopter and 6 for the ship's deck, with these latter being dependent on the influence of the sea waves on the ship's behavior). A second difficulty is related to the task's demands (e.g. accuracy of ± 1.5–2 m in position and ± 5° in azimuth required [1] to land on the 14.2 m wide deck of the Lafayette class Frigate). Finally, the weather conditions can critically affect the pilots' information pickup processes, for instance when seawater sprays the helicopter's windscreen [2]. Controlling landing maneuver is so difficult for pilots that it has motivated the

github.com/AntoineHPMORICE/
HelicopterShipLanding-TauDot_data.

**Funding:** The authors received no specific funding for this work.

**Competing interests:** The authors have declared that no competing interests exist.

development of automatic control mechanisms to prevent fatalities. Vision-based feedback control systems have been proposed to control automatically the approach phase [3] and optic flow-based automatic decking has also been successfully tested with on-board small aerial robots in lab conditions [4]. With those solutions, pilots are not inside the loop, limiting their effectiveness in case of complex situations and introducing issues when pilots must regain control. In this article, we want to study an alternative solution to prevent fatalities, avoiding the listed issue of automatic control by providing additional visual information aimed at improving pilots' performance and safety during ship landing situations.

The landing task is mainly, visually regulated with natural cues which are provided by different information sources as the helicopter approaches the deck (see [5] for task analysis) but is also assisted by additional visual information provided by some existing shipboard aid systems–whenever equipped. Firstly, the *Horizontal Reference Bar*, a lighting system fixed on the back of the superstructure that remains horizontal and thereby helps pilots to perceive the ship's roll. Secondly, the *Glide Slope Indicator*, a tricolor beam, helps pilots to visually establish and maintain the proper descent slope for a safer landing. However, despite such aids, deck landing remains a difficult and risky maneuver. Complementary visual aids, feeding the pilots' natural information-movement coupling, must therefore be designed.

## 1.1 Designing interfaces for assisting helicopter ship landing

The objective of designing visual assistance adapted to the users' needs is intimately linked to aeronautics development [6, 7]. In the early stages of development, the human factor issues involved in their design were raised [8]. Research was first carried out on the best symbology to display the attitude of fighter planes on Head-Up-Displays [9]. More complex symbologies depicting flight path later demonstrated their relevance for upgrading flight path guidance and reducing workload [10]. However, display modality is only one amongst many ergonomic considerations, and relevant information also had to be provided to users to improve performance. Task analysis was used, initially and continuously to investigate pilots' habits in picking up cues and regulating their maneuvers [5, 11, 12]. Such methodologies provide insight about the available, relevant, and used perceptual variables and perceptual-motor strategies when landing. However, perceptual-motor processes may not reach the pilots' awareness. Virtual reality setups thus allow more finely grained experimental methods that can be used in complement to track perceptual information picked up when landing [13]. Indeed, the more rooted the aid is in the perceptual-motor principles used by pilots, the more efficient the aid will be [14].

The framework of *Ecological Interfaces Design* ([15] hereinafter referred to as EID) aims to tackle the problem of identifying the suitable information able to improve operator's performance through a two-step approach [16–18]. The first step consists of analyzing the work domain, here defined as the pilot-helicopter-ship system with the ship's deck motion acting as a forcing function on the pilot-helicopter system, to create, in the second step, an augmented reality interface that would make the crux variables visible allowing pilots to directly control the system.

**1.1.1 What to display?.** At the first step, EID prescribes that to determine "*what to display*" [18], the complexity of the socio-technical work domain must be described to reveal the relevant *structure and content* of the work domain [17, 18]. The abstraction hierarchy [19] is the elicited tool for this purpose. We illustrate in Fig 1B the *Content* of the pilot-helicopter-ship system during the deck landing task as a traditional five levels of constraints' class.

At the first level, where the goals of the system are defined, two *functional purposes* can be identified. The first postulates that, in both civil and military contexts, fuel should be

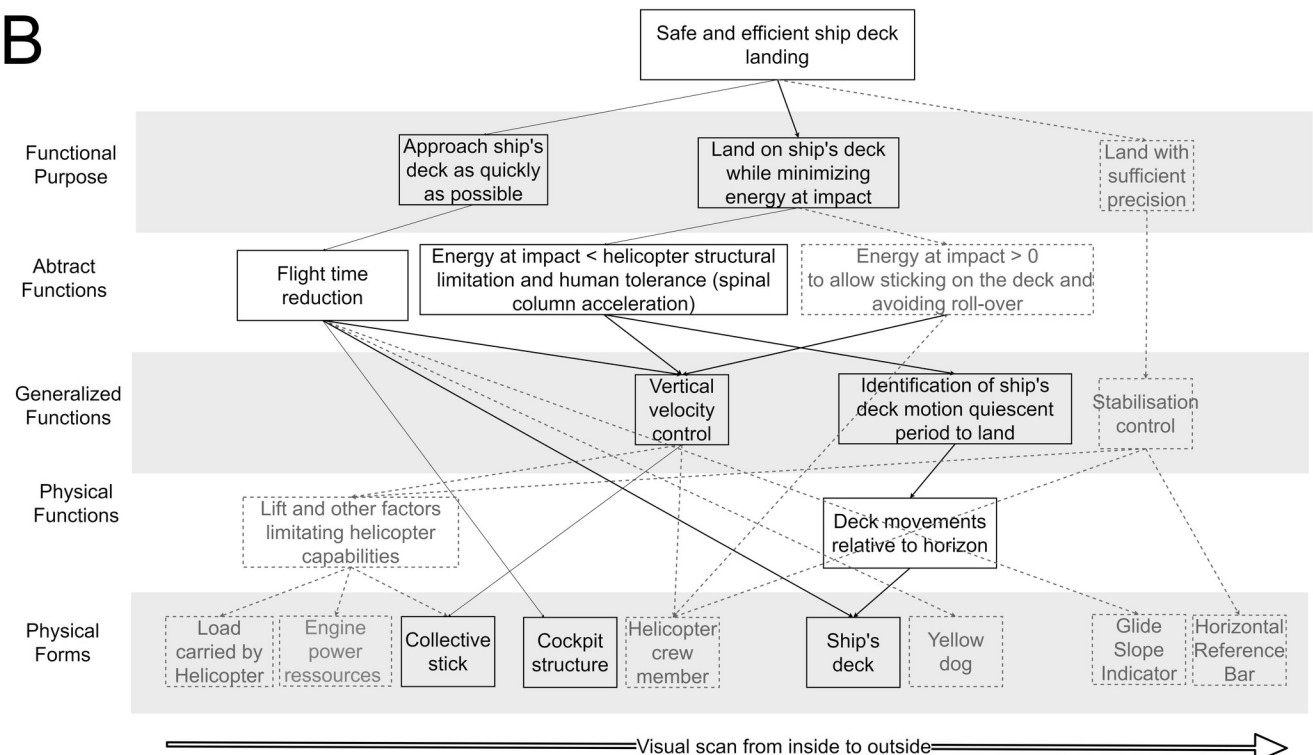

**Fig 1.** (A) Illustration of the constraints acting on the visual system of pilots during landing on the deck of a ship. (B) Abstraction hierarchy for the helicopter deck landing task. Dotted lines indicate some additional variables to be considered for real-life applications.

economized and time to reach the ship's deck minimized during the approach phase. The second, highlighted by the work of Thomas *et al.* [20], is to minimize the energy at impact during the landing phase. The author evidenced that expert pilots attempting to perform deck landing on a realistic fixed-base helicopter simulator, attempt to minimize energy at impact when landing on-sight by strongly coupling the altitude of their helicopter with vertical deck motions from the hover position. Note that the task's accuracy demands should also be processed despite not being considered in this study. At the second level, the *abstract function* aims to describe the causality links governing the purposes of the system. During the approach phase, the picking up of sources of visual information about the heave motion of the ship's deck becomes more and more complicated as the latter becomes increasingly occluded by the cockpit. The outside-the-cockpit information pickup process is complicated by the mere presence of the cockpit structure that reduces the pilots' downward field of view (FOV). *MIL-STD-850B* [21] states that the downward outside-the-cockpit FOV can be limited to −25˚ at 0˚ azimuth and up to −50˚ below the pilots' eye level from 10˚ to 135˚ azimuth, respectively. Fig 1A illustrates that such FOV occlusion induces pilots to preferentially use the door windows (and chin windows when available) during approaches and landings as reported by [22] and to rely on other crew members who look through the side door and provide information to the pilot. Information provided by yellow dog on the flight deck via radio communication [23] imposes an additional load on the pilot who limits or even generally cuts off the communication with the ship's crew in the last phase before the landing. This situation makes the reduction of horizontal and vertical FOV [24] detrimental to rotorcraft control. Therefore, several approaches have been proposed to overcome FOV-related problems such as adjustment of the pilot's vertical seating position [25] and several other solutions (redesign of glare shield, chin window, mirror, cockpit visual enhancements, etc.) [22] that offer low-cost solutions with a theoretical substantial gain, but which can be of limited effectiveness in complex military conditions. During the landing phase, the prescribed touchdown energy is limited because passing this limit would cause structural damage to the helicopter [26] and trauma to the pilots' spines [27] but it should also be sufficiently positive to avoid helicopter roll-over. At the third level, *generalized functions* that allow the aforementioned abstract functions are described. Putting aside the control of the stability of the helicopter, the conflict between information pickup and minimization of energy at impact is controlled by the regulation of helicopter vertical velocity. The energy minimization problem is related to the ship's motion quiescent period. At the fourth level, the *physical function* describes the properties of the components used to drive the aforementioned function. In real-life, vertical velocity is regulated through the control of lift [5], an action capability that is bounded by helicopter engine power, load carried, etc. We considered a simplified helicopter in which the vertical velocity depends only on the collective stick. At the fifth and bottom level, the appearance and location of the helicopter with respect to the ship's deck, the collective stick controlling the changes in altitude along the longitudinal axis are described. The aforementioned *Glide Slope Indicator* and *Horizontal Reference Bar* help for the approach and landing phases, respectively.

The abstraction hierarchy is more than a stratified hierarchical description of the workspace, it also allows the latter's structure to be defined through means-end relationships linking the adjacent levels in a 'why-what-how' relation [16]. These links describe available means for achieving goals, thus specifying by affordances [28] how the approach and landing phases can be carried out. When performing a sight landing, both purposes of flight time reduction and

energy minimization at impact are achieved by picking up from optic flow on the ship's deck (see Fig 1A) one among several candidate optical variables to visually control the decrease of the rotorcraft's approach velocity towards a null value when impacting the ship's deck. Note that pilots aim to land with a slightly positive velocity relative to the ship's deck in order to keep the helicopter on the deck and avoid roll-over. An overview of these optical variables and corresponding strategies (see "Braking" section in [29] for details) can be summed up as follows. First, pilots would use the $\dot{\tau}$ variable (see [30] for the seminal hypothesis), the first temporal derivative of $\tau$. In the case of helicopter ship's deck landing, $\tau$ denotes the current visual angle of the deck ($\theta$) divided by its rate of expansion ($\dot{\theta}$) and specify the time-to-contact with the deck if the helicopter is moving at constant speed. Maintaining tau-dot ($\dot{\tau}$) at a value equal to −0.5 is a minimalist but an efficient perceptual-motor strategy for visual control of braking when driving [31], or for visual control of locomotion when decelerating to grab a door handle [32]. Its use for visual control of flight concerns as much pigeons landing on a perch [33] and hummingbirds approaching a feeder [34] as helicopter pilots performing stopping maneuvers on the DERA Advanced Flight Simulator in Liverpool [35]. While the use of $\dot{\tau}$ looks to be mandatory to successfully perform the landing task, its pickup should be enhanced since only experts may be sensitive to such a "high-order variables". Novices [36], or children [32], might pick up less relevant, "low-order", variables. Moreover, even when $\tau$ information is available, observers often fail to use it properly ([37], chapter 1). Therefore, as stated by Padfield [35], $\dot{\tau}$ should be the key variable to guide the design of vision augmentation systems. Alternatively, pilots can pick up the rate of expansion of the current visual angle of the deck ($\dot{\theta}$) to maintain it at a constant positive value ([38, 39]). This strategy was hypothesized to trigger the initiation of braking but the regulation of braking with respect to these variables was not demonstrated [29]. Besides, maintaining constant the rate of the expansion of texture elements was also reported as a perceptual-motor strategy used by honeybees to decelerate when landing (see [40], generalized later in [41], and [42] for a review), giving support to the model of constant-$\tau$ guidance for landing. Later, the $\tau$ variable was proposed as a way of enslaving automatic control of landing of a helicopter on a ship [43]. When observing playback of the visual scene captured during automatic control of landing, pilots judged the landing maneuver natural, still arguing for a $\tau$ based regulation of landing by pilots.

In sum, the constraints acting on the visual system of pilots during landing on the deck of a ship are summarized on Fig 1A. Several perceptual variables are candidate to help pilots regulating the helicopter's descent: $\tau$, $\dot{\tau}$ and $\dot{\theta}$. These variables must be extracted from the optic flow generated by the ship's deck but are occluded by the helicopter's cockpit.

**1.1.2 How to display?.** At the second step, EID prescribes that to achieve the "*how to display*" [18], the display must act as a smart perceptual instrument, exploiting the power of direct perception [17], to convey or communicate in an effective way [16] higher-order information to the operator through a relevant *form* of the interface [17]. A smart way to communicate required steering and velocity corrections to rally drivers while cornering is to project in the HUD ideal, limit and future trajectories [17]. In the same vein, 3D projections of the current total energy of the aircraft with respect to the ideally targeted energy can inform aircraft pilots about how to manage their vertical acceleration [44].

To cope with this occlusion problem during the final maneuver and with the tau-based strategy, two visual augmentations are respectively proposed.

With a *Replication* of the deck, usually occluded by the cockpit during the final part of the maneuver, the pilots' visual range can be extended by augmented reality technology beyond the cockpit occlusion. Such a paradigm is also called "seeing into the walls" [45]. Such a visual augmentation would allow operators to regulate their maneuver to its end by picking up all the

optical variables candidates for regulating the maneuver $\tau$, ($\dot{\tau}$ and $\dot{\theta}$). A such augmentation provides an original solution to overcome FOV-related problems. However, it does not convey the key relationship between the helicopter-ship system in an efficient manner, as it is still possible to extract several perceptual variables to visually control the descent.

With an *Addition* in a gauge of a synthetic $\dot{\tau}$-related information, whose pickup in the real environment may be complicated by the complexity of military operation or by the lack of attunement of operator to it, the pilots can have a direct reading of the relationship between their current value of $\dot{\tau}$ and the ideal $\dot{\tau}$ value of -0.5 which provides higher-order information concerning whether pilots are executing a soft landing and, if not, how to correct. Tactile displays designed in the ecological framework [46] have demonstrated their efficiency to convey higher-order information akin to time to contact. Displaying the current vs. ideal $\dot{\tau}$ could take the form of a moving pointer in a gauge. Such an interface was successfully helping operators to online regulate their locomotion behavior with respect to an ideal value [47]. In the present study such an interface may thus allow operators to regulate their approach velocity by decelerating smoothly in order to minimize energy at impact when impacting the ship's deck. Therefore, the Addition augmentation may replace the expertise of the operators in the $\dot{\tau}$ pickup.

### 1.2 Aim of the present study

This study explores the nature of the additional visual augmentation that could improve helicopter landing behavior. We firstly considered a visual augmentation design that was based on in-field analysis and consisted of solving cockpit-induced visual occlusion problems. We hypothesized that pilots' performance could be improved if sources of information carried by the ship's deck were fed to the pilots while the deck was occluded by the helicopter cockpit during the final part of the landing maneuver. We thus tested an augmentation consisting of a *Replication* of the visual ship deck scene (including $\dot{\tau}$, but also other optical source of information). An improvement of performance in this *Replication* augmentation would suggest that pilots are able to visually couple with any optical information available, but they are hampered by cockpit occlusion in control condition. We secondly considered a visual augmentation design that was grounded in ecological psychology and consisted of strengthening the online regulation of the deceleration by keeping the current $\dot{\tau}$ variable around the $\dot{\tau} = -0.5$ ideal value. We hypothesized that pilots' performance can be improved by the *Addition* of a gauge allowing a direct reading of the current value of the $\dot{\tau}$ variable in comparison to the ideal $\dot{\tau}$ value. Indeed, that relationship is a higher-order variable that informs pilots' about whether they are executing a smooth and efficient deceleration leading to a soft landing, a too soft landing (i.e., stopping short of the landing point), or a too hard landing (i.e., landing on the deck with a velocity at impact exceeding helicopter structural limitations and spinal column tolerance). A performance improvement in this *Addition* augmentation would suggest that, not only the information-movement coupling can be fed while information pickup is interrupted by occlusion, but also that it is important to feed it with $\dot{\tau}$ variable for improving performance. We report our methods and analyses gained with a fixed-base helicopter simulator in the following sections.

## 2. Methods

### 2.1 Population

16 participants (13 men and 3 women, aged 24.7±2.9 years) volunteered for this experiment. Participants were recruited from among students at the ONERA center (Salon-de-Provence, France) and the Faculty of Sport Sciences (Marseille, France) who had responded favorably to a volunteer search advertisement. To apply, participants had to be right-handed, not play

video games, have normal or corrected vision, and have no experience in aircraft piloting. The sample size was determined according to [48] and was thus considered as representative of young adults who were healthy but novices in helicopter piloting. Participants were not informed about the purpose of this study but were informed about the experimental procedure, which was approved by the local committee, and signed a consent form following the requirements of the Declaration of Helsinki. The experiment was run in April 2018 in the Department of Information Processing and Systems at ONERA (Salon-de-Provence, France).

## 2.2 Apparatus

Fig 2 shows the *PycsHel* fixed-base rotorcraft engineering simulator of the Department of Information Processing and Systems at ONERA (Salon-de-Provence, France) we used. Participants were seated on the right-hand side of a typical helicopter cockpit with side-by-side seat configuration. The seats were placed in front of 3 vertical large screens (3.16 m wide × 2.37 m height) perpendicularly arranged and a large horizontal screen, which encompassed 265° of their horizontal and 135° of their vertical field of view. The virtual scene was projected onto the screens using four identical DLP video-projectors (W1080ST+, BenQ™, Taipei, Taiwan) each having a resolution of 1920 by 1080 pixels, refreshed at 60 Hz.

## 2.3 Visual world

From the participants' viewpoint, the visual scene was composed of an immobile sky with clouds, a dynamic sea with waves that influenced the heave movements of a 125-m long,

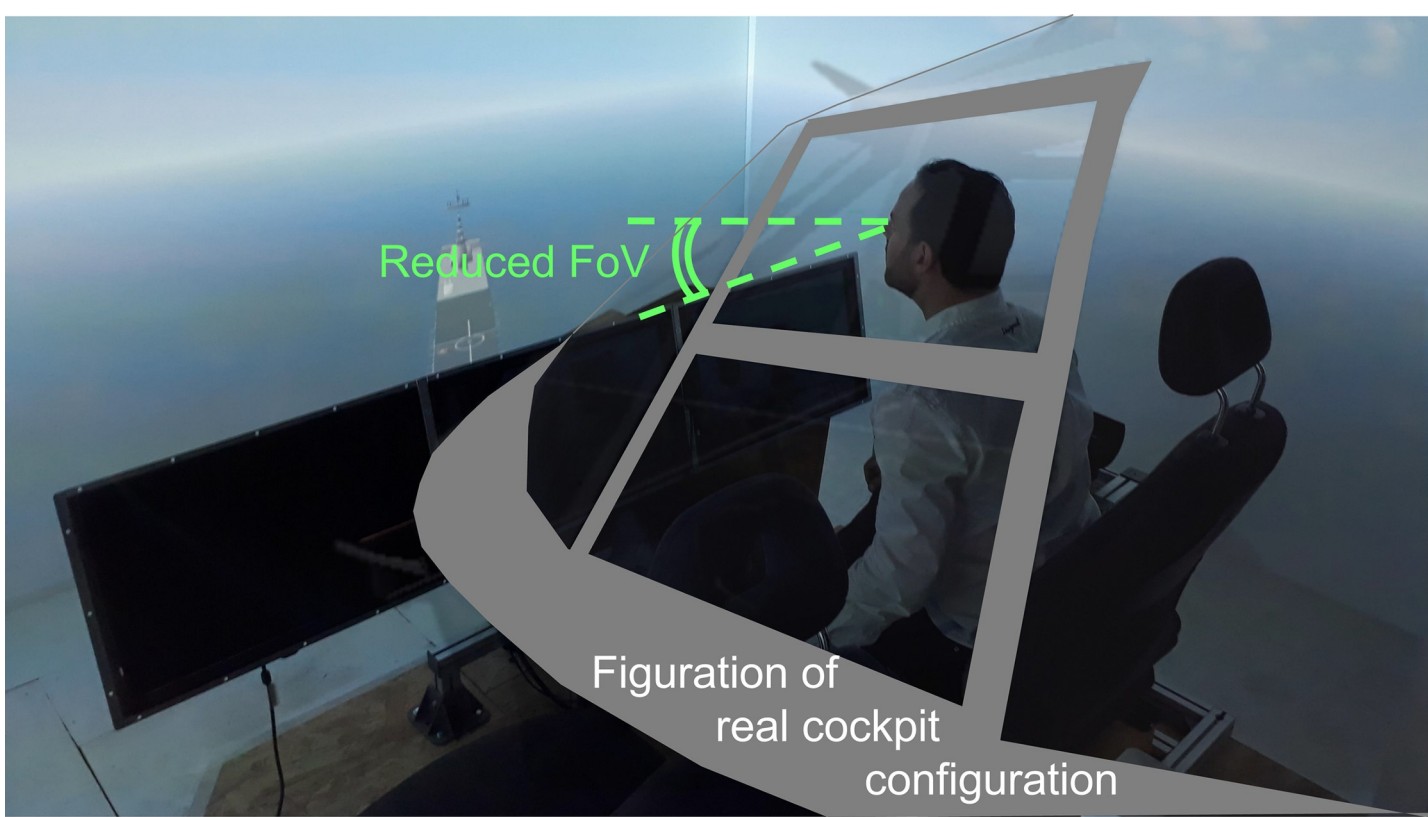

**Fig 2. In the simulator, a set of three, switched off, LCD monitors in front of the operators reproduce the occlusion of the pilot's vertical FOV of an actual rotorcraft cockpit.** Users can continue picking up information on both sides of the cockpit. The visual scene is enslaved to the virtual helicopter displacement and is displayed inside a CAVE.

15.40-m wide and 40-m air draught (*Lafayette* type) frigate with a 15.42-m radius deck platform. The sea motion did not influence the ship's surge, sway, roll, pitch or yaw angular motions. The frigate was not moving along its longitudinal axis.

## 2.4 Task

The task consisted of visually controlling, without instruments, the landing of a virtual 10-ton class, cargo type helicopter on the deck platform located at the stern (rear) of a frigate class ship at sea. The helicopter landing maneuver was performed along a "12 o'clock" direction, called "astern approach", as it is currently done in the French naval forces, where the rotorcraft follows the ship along its longitudinal axis. As the participants lacked piloting experience, the helicopter motion and commands were simplified with regards to real conditions by disabling all rotational (roll, pitch, and yaw) and lateral movements, for the helicopter as well as for the ship. As a result, the helicopter trajectory is purely longitudinal. This would allow novices participants to focus on the coupling between the longitudinal (i.e., forward, backward) movements of the helicopter and the visual sources of information emanating in return from the environment. Moreover, the helicopter mass center is constrained to move on a pre-computed trajectory within the vertical plane, that was modeled from previous records of expert pilots landing in the simulator [49]. This trajectory started at 90 m behind the deck center point, and 14.64 m above the deck level and ended as soon as the rotorcraft landing skids were in contact with the deck platform. Therefore, the trajectory guides the helicopter to land at the center of the deck so that participants were thus unconcerned either with landing accuracy with reference to the deck platform or the regulation of the rotorcraft's attitude.

Participants were instructed to adjust with their left hand the position of the collective stick that regulated, through second-order dynamics, the speed of the rotorcraft (i.e., pulling, pushing and standby actions on the stick induced deceleration, acceleration and constant rotorcraft speed, respectively) to minimize the duration of the maneuver while also minimizing the energy at the moment of the impact with the deck platform.

## 2.5 Independent variables

We explored the nature of the additional visual information that could improve helicopter pilots' landing behavior. We also investigated whether sea state influenced the usefulness of additional information. Two variables were thus manipulated within-subject (*Sea*: 2 modalities, and *Augmentation*: 3 modalities). Fig 3 depicts the typical visual scenes during these manipulations.

The *Sea* manipulation was designed to manipulate the level of difficulty of the task between two modalities (*Calm sea*, *Rough sea*, see Fig 3, top panels for screenshots). In *Calm sea*, the water surface was flat and the ship's heave amplitude was equal to 0. In *Rough sea*, the waves movements influenced the ship's heave (see [50] for details about the relationship between sea state and ship deck motion). The wave dynamics are defined by a sea state equal to 5 on the Douglas scale, with a heave motion amplitude reaching 3.3 m—knowing that in real operational conditions, a significant ratio of qualified helicopter pilots will not perform ship landings in a sea state equal to or greater than 5–6.

The *Augmentation* manipulation was designed to provide additional informational content (*Replication* and *Addition*) overlaid on the natural visual scene (*Control* condition). In the *Control* condition, the visual scene content simulated that of a natural scene and the landing approach was regulated only by sight.

In the ship's deck *Replication* modality, the scene was enriched to compensate for the occlusion of the deck platform by the helicopter cockpit. In our setup, the outside view was occluded by the cockpit below an angle of $-14.13°$ under the horizon. Therefore, the visibility of the ship's deck

## Calm sea

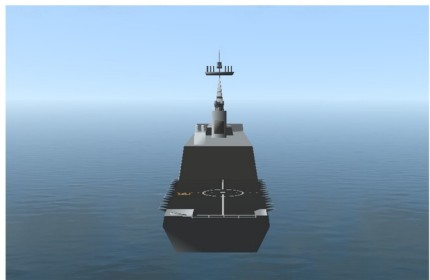

## Rough sea

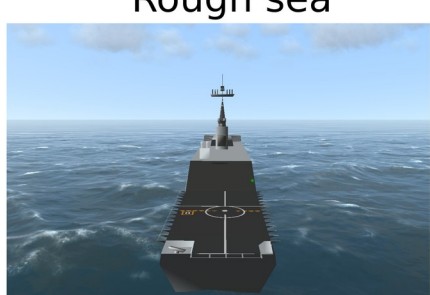

## Control

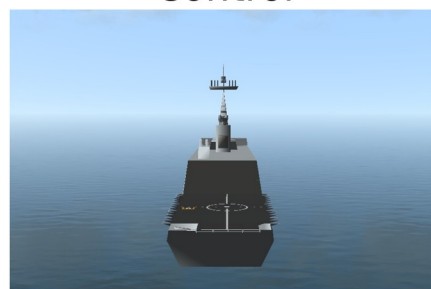

## Deck Replication

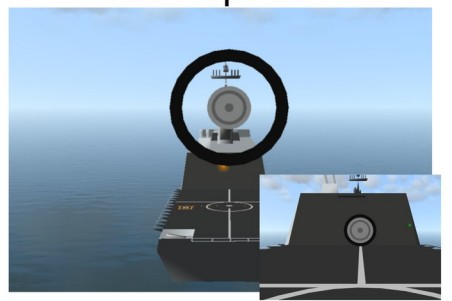

## Addition of $\dot{\tau}$

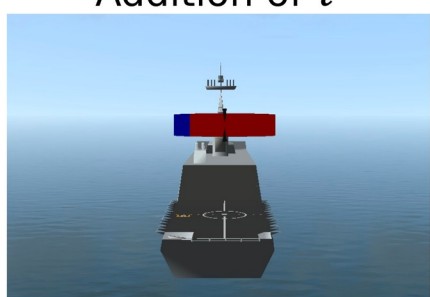

**Fig 3.** Typical visual scenes depending on the *Sea* (top panels, 2 modalities) and the *Augmentation* manipulations (bottom panels, 3 modalities).

was first partially and then fully reduced during 76% and 27% respectively of the traveled distance (see Fig 4A for a schema of visual occlusion). For this reason, a black ring, appearing as fixed with respect to the helicopter frame and a grey disc with white markings, identical to those of the ship deck platform were overlaid on the *Control* condition's scene. The grey disc moves along the horizontal axis with a motion pattern homothetic to the relative vertical distance between the helicopter and the deck. When the helicopter touches the deck, the disc will fill the inside diameter of the ring. In this sense, the *Replication* mimics a bird's-eye view of the deck platform enslaved to the current altitude of the rotorcraft, as seen in Fig 3, bottom middle panel.

In the $\dot{\tau}$ *Addition* modality, the scene was enriched to help participants to regulate their landing behavior by canceling the difference between the current $\dot{\tau}$ value and the ideal $\dot{\tau}$ value ($\dot{\tau}$ = −0.5) and thus minimizing kinematic energy at impact. This augmentation behaves like a moving scale gauge along with a fixed (in the helicopter frame) pointer as illustrated in Fig 3, bottom right panel. The scale consists of red and blue areas, corresponding to current values of $\dot{\tau}$>−0.5 and current values of $\dot{\tau}$<−0.5, respectively. Therefore, when the cursor is perfectly aligned with the delimitation between red and blue zones, the current $\dot{\tau}$ value is equal to −0.5 and the current participant's behavior makes the rotorcraft decelerate such that kinematic energy at impact will be null. If the participants land while the cursor is located in the red zone, then $\dot{\tau}$>−0.5 and the impact energy will be too high. If the participants land while the cursor is located in the blue zone, then $\dot{\tau}$<−0.5 and the helicopter will stop before reaching the landing platform.

The current value of $\tau$ is computed in real-time using kinematic variables available from the simulation, as in (1):

$$\frac{1}{\tau} = \frac{|\vec{V}.\vec{X}|}{\vec{X}.\vec{X}} \tag{1}$$

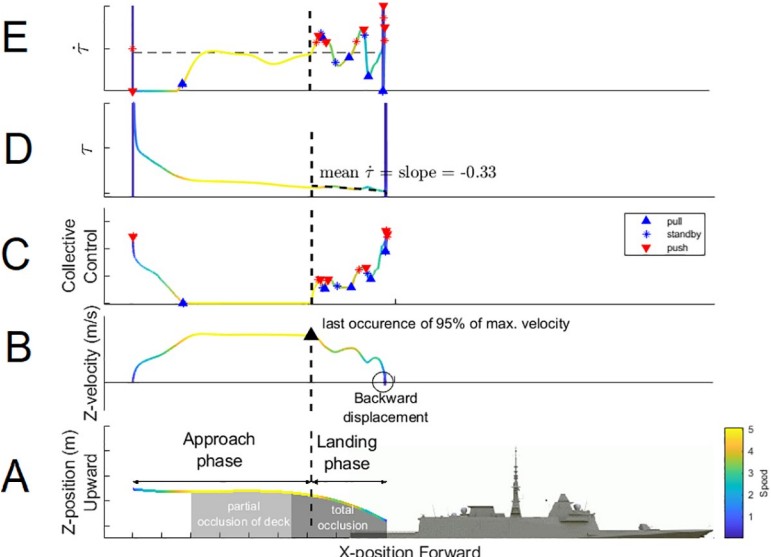

**Fig 4. Dependent variables extracted during each sample trial.** (A) The dark and light gray areas depict partial and total occlusion of the ship's deck by the cockpit during the maneuver. (B) Helicopter velocity is depicted by colors. The approach phase ended, and the landing phase started when the helicopter velocity started to decrease (95% of max). (C) The changes performed in the collective stick during the trial comprised pull (i.e., increase velocity, ▲ symbols), standby (i.e., keep velocity constant, * symbols) and push actions on collective stick (i.e., decrease velocity, ▼ symbols). (D) The $\dot{\tau}$ strategy consisted of maintaining the slope of $\tau$ around −0.5, that produced (E) oscillations around $\dot{\tau}$ equal to -0.5.

where $\overrightarrow{X}$ and $\overrightarrow{V}$ are respectively the position and velocity arrays of the helicopter with respect to the ship's deck reference frame.

$\dot{\tau}$ is calculated as the time derivative of (1).

Moreover, to enhance the sensitivity of the gauge displacement in the vicinity of the target zone around $\dot{\tau}$ equal to −0.5, a nonlinear mapping function has been designed between the current value of $\dot{\tau}$ and the gauge position using a symmetrized square-root function.

Both augmentation display zones are immobile in the visual scene and projected onto the virtual scene at the same dimensions (virtual object of 37 cm in diameter displayed 100 cm from the subject) to be observable with an optical angle of $20^{\circ}$, sufficient to discriminate shapes and colors.

## 2.6 Protocol

Before the experiment, the participant read the instructions that were then repeated orally by the experimenter. During a Familiarization phase, a minimum of one practice run for the two *Sea* and the three *Augmentation* modalities was then provided for each participant. An additional practice trial was allowed depending on the participant's understanding of the experimental conditions. A posteriori analyses ensured the familiarization phase was long enough to allow participants to calibrate themselves with the task and *Augmentations* (see *Control of perceptual learning during the Familiarization phase* section in S1 Fig in S1 File). The experiment phase was organized into 6 sessions of 7 trials, corresponding to the 6 combinations of *Sea* and *Augmentation* modalities, respectively. The order of the trials was randomized for each participant. After each session, participants were required to complete the Modified Cooper-Harper Handling Qualities Rating Scale (cf. *Dependent Variables* section). A short rest was permitted between sessions if requested.

Each experimental trial was initiated manually by the experimenter and started when the collective stick automatically recovered its neutral position, using a motorized trim. Each trial lasted 90 sec. maximum. The full experiment lasted 105 minutes.

## 2.7 Dependent variables

The potential benefits, as well as detrimental effects of augmented reality, must be scrutinized with different, possibly interconnected, levels of analysis. Indeed, additional displays have already been found to influence operator's workload [51], task performance [52], and information-based strategy [53]. We thus targeted the following dependent variables to reveal the influence of experimental manipulation at those levels of analysis.

**2.7.1 Workload level.** Participants' perception of the difficulty of the landing task was assessed through the measure of Mental load, which was retrieved via the Modified Cooper-Harper Handling Qualities Rating scale [54] completed by participants after each session block (i.e., after each block of 7 trials combining an *Sea* and an *Augmentation* modality). Cooper-Harper ratings were converted into Z-scores before analysis.

**2.7.2 Performance level.** Participants' compliance with instructions was assessed using the following performance indicators in the landing task criteria.

*2.7.2.1 Energy at impact.* The kinematic energy at impact ($E$, in J) was computed according to (2):

$$E = \frac{1}{2} \times m_{heli} \times |\overrightarrow{V_{impact}}|^2 \qquad (2)$$

with $m_{heli}$ the mass of the helicopter set at 8000 kg, and $|\overrightarrow{V_{impact}}|$ the speed of the helicopter relative to ship at the moment of impact.

*2.7.2.2 Duration of maneuver.* The total duration of the maneuver (in sec.) was computed as the time elapsed between the trial start and touchdown. The approach phase ended, and the landing phase started when the helicopter's velocity started to decrease (from 95% of max). The duration of the landing phase (in sec.) was computed as the time elapsed from the last occurrence of 95% of the maximal velocity (i.e., the moment of the first deceleration) until touchdown.

*2.7.2.3 Helicopter phase at impact with respect to the ship's heave cycle.* The heave cycle of the ship was defined as the ship motion between two maximum vertical positions. For each heave cycle, ship velocity and position values were centered, normalized and interpolated into 360 bins. Within trial average phase plane (ship velocity as a function of ship position) were thus computed and averaged across participants. With this definition, optimal helicopter phase at impact in the ship's heave cycle should occur at a little more than 180° since pilots would aim to land with a slightly positive velocity relative to the ship's deck in order to keep the helicopter on the deck and avoid roll-over the ship. The helicopter phase at impact ($\varphi$) in the ship's heave cycle were thus computed according to (3), as the phase angle (in deg.) of touchdown in the phase plane, and averaged across participants (see also [55] for another implementation of this method).

$$\varphi = arctan(Ship\ Vertical\ Velocity_{touchdown}/Ship\ Vertical\ Position_{touchdown}) \qquad (3)$$

with this convention, $\varphi$ was equal to 0° at the maximal ship position and equal to 180° at the minimal ship position.

**2.7.3 Thrust commands level.** Participants' command of the rotorcraft engine was assessed through the computation of the variables summarized in Fig 4A–4C.

*2.7.3.1 Acceleration.* The velocity signal was retrieved from simulator outputs and filtered with a zero-phase forward and reverse digital low pass Butterworth filter (cut-off frequency: 8

Hz, order: 4). The acceleration signal was computed as the first derivative of the filtered velocity signal. We extracted from the acceleration signal the values of the average and maximum deceleration (in $m/sec.^2$) during the landing phase.

*2.7.3.2 Backward displacements.* We monitored backward displacements by separately computing, from negative value parts of the filtered velocity signal, the cumulative duration (in sec.) and the number of occurrences of backward displacement.

*2.7.3.3 Collective stick reversal.* Collective stick signal was retrieved from the simulator outputs. We extracted the number of occurrences of each collective stick reversal.

**2.7.4 Perceptual-motor strategy level.** Participants' information-based strategy for visual landing regulation is available through the calculation of the critical $\dot{\tau}$ that initiates the collective stick adjustments. Indeed, since $\dot{\tau}$−based perceptual-motor strategy is assumed to guide action (i.e., whether to slow down, speed up or keep speed constant), the pull, push and standby collective stick actions would be initiated at specific, relevant current values of $\dot{\tau}$. We therefore extract the value of $\dot{\tau}$ during push, pull, and standby actions on the collective stick (see Fig 4C–4E) and compare their cumulative distribution with the ideal value $\dot{\tau}$ of -0.5 that will highlight the signature of minimalist perceptual-motor regulation of landing (as shown by [31]).

## 2.8 Statistics

Our first aim was to investigate the combined influence of *Sea* and *Augmentation* manipulations on each level of analysis. We therefore performed, for all dependent variables, 2-way analyses of variance with repeated measures, (2-way RM-ANOVA) on the two *Seas* (*Calm sea*, *Rough sea*) and on the three *Augmentations* (*Control*, *Replication*, *Addition*). For all ANOVAs, partial effect sizes were computed ($\eta_p^2$) and post-hoc tests were conducted using Tukey HSD a posteriori tests in case of significant main effect and/or interaction between *Sea* and *Augmentation* factors to evidence significant differences between modalities.

We secondly compared the benefits of the *Addition* with those of the *Control* condition. Since we had hypothesized that augmented reality is most useful when the ship is being thrown around by a *Rough sea* and that a maximum improvement of landing behavior will be allowed by the ecologically grounded augmentation (i.e., *Addition*), we performed unilateral paired t-tests to evaluate the benefits offered in *Rough sea* by the *Addition* in comparison to the *Control* modality.

## 3. Results

### 3.1 Workload level: Do augmentations lighten the mental load?

This first section was designed to study whether augmentations could lighten mental load. Moreover, we investigated how the sea's state influenced this for participants. We predicted an increase of mental load in *Rough* in comparison to *Calm* sea since participants were additionally required to cope with ship heave movements when minimizing energy at impact. We also predicted that both designs of augmentation would facilitate the participants' task (i) since visual information was available during the entire maneuver and participants were not required to move their heads to pick up information (*Replication*) and (ii) since abstracted relevant information was available throughout the maneuver via the gauge (*Addition*).

A 2-way RM-ANOVA performed on the individual average Cooper-Harper ratings (also see S2 Fig in S1 File) revealed a significant main effect of the *Sea* factor (F(1,13) = 105.62, p<0.001, $\eta_p^2 = 0.89$). The Cooper-Harper ratings in *Calm* sea were significantly lower than those obtained in *Rough sea* (1.64±0.15 vs. 4.55±0.27, p<0.05). This confirms that the heave movements in *Rough sea* increased the perceived difficulty of the task. The ANOVA also

yielded a significant effect of the *Augmentation* factor (F(2,26) = 4.87, p<0.05, $\eta_p^2 = 0.27$) but no significant *Sea×Augmentation* interaction (F(2,26) = 1.5, p>0.05, $\eta_p^2 = 0.10$). Post-hoc tests revealed that the Cooper-Harper ratings gained with the *Replication* were significantly lower than those obtained in the *Addition* modality (2.57±0.16 vs. 3.46±0.32, p<0.05). To summarize, the mental load increased with the sea state and decreased with the *Replication* compared to the *Addition* modality. Counter-intuitively, the mental load does not decrease with the *Addition*.

### 3.2 Performance level: Do augmentations improve landing performance?

This second section investigated the effect of *Sea* manipulations on participants' landing performance and whether the augmentations improved it. We hypothesized that the benefits of the augmentations can be seen on energy at impact, duration of maneuver, and relative phase on the touchdown with respect to the ship's heave motion.

**3.2.1 Energy at impact.** The energy at impact would reflect how well participants followed the instruction to minimize it. Fig 5A shows that the energy at impact clearly increased in *Rough* sea in comparison with *Calm* sea. A slight decrease of the energy at impact was observed in *Replication* and *Addition* with respect to the Control modality in *Rough Sea* only. A 2-way RM-ANOVA performed on the individual average values of energy at impact revealed a

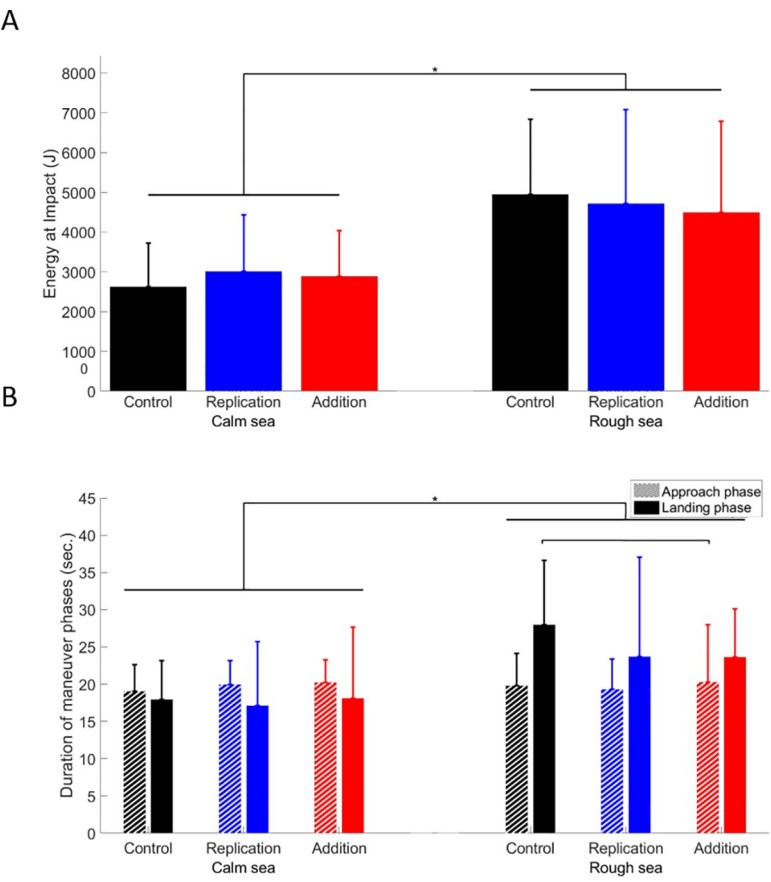

**Fig 5. Inter-individual average values of performance variables.** (A) Kinematic energy at impact (J), and (B) Durations of the approach and landing phases (sec.). Vertical bars on histograms depict the standard deviation of individual average values.

significant main effect of the *Sea* factor ($F_{(1,15)} = 17.02$, $p < 0.001$, $\eta_p^2 = 0.53$), but no significant effect of the *Augmentation* factor ($F_{(2,30)} = 0.12$, $p > 0.05$, $\eta_p^2 = 0.01$) nor any significant effect of the *Sea×Augmentation* interaction ($F_{(2,30)} = 0.85$, $p > 0.05$, $\eta_p^2 = 0.05$). The energy at impact was significantly increased in *Rough* in comparison to *Calm sea* (4721±765 vs. 2842 ±427 *J*, $p < 0.05$), probably because of the participants' difficulty in fully compensating for the ship heave movements. The *Augmentation* manipulation does not significantly improve the participants' ability to minimize energy at impact.

**3.2.2 Duration of maneuver.** The duration of the maneuver would reflect how participants managed to save time when completing the maneuver. A 2-way RM-ANOVA performed on the individual average values of the total duration of the maneuver revealed a significant main effect of the *Sea* factor ($F_{(1,15)} = 13.12$, $p < 0.05$, $\eta_p^2 = 0.47$), but neither any significant main effect of the *Augmentation* factor ($F_{(2,30)} = 0.89$, $p > 0.05$, $\eta_p^2 = 0.06$), nor significant *Sea×Augmentation* interaction ($F_{(2,30)} = 1.20$, $p > 0.05$, $\eta_p^2 = 0.07$). The total duration of the maneuver was significantly higher in *Rough* than in *Calm sea* (44.91±2.57 vs. 37.45±1.55 sec., $p < 0.05$). A paired t-test conducted on the individual average values of the total duration of the maneuver to investigate a potential overall time-saving gain in *Rough sea* between the *Control* and *Addition* modalities indicated a decrease, nearing significance, in the total duration of the maneuver (47.8±9.6 vs. 43.9±10.5 sec., $t_{(15)} = 1.59$, $p = 0.07$) between the *Control* and *Addition* modalities in *Rough sea*.

To investigate further the part of the maneuver that was influenced by *Sea* and *Augmentation* factors we thus distinguished, as shown on Fig 5B, the approach phase (the beginning of the trial until the first deceleration) from the landing phase (first deceleration to touchdown). 2-way RM-ANOVAs were separately performed on the individual average values of the duration of the approach phase and the duration of the landing phase. The 2-way RM-ANOVA performed on the duration of the approach phase did not revealed neither any significant main effect of the *Sea* factor ($F_{(1,15)} = 0.005$, $p > 0.05$, $\eta_p^2 < 0.001$) or of the *Augmentation* factor ($F_{(2,30)} = 0.76$, $p > 0.05$, $\eta_p^2 = 0.05$), nor a significant *Sea×Augmentation* interaction ($F_{(2,30)} = 0.55$, $p > 0.05$, $\eta_p^2 = 0.03$). Therefore, the approach phase duration remained constant at around 19.76 sec. whatever the experimental manipulations. Conversely, the 2-way RM-ANOVA performed on the individual average values of the duration of the landing phase revealed a significant main effect of the *Sea* factor ($F_{(1,30)} = 15.75$, $p < 0.05$, $\eta_p^2 = 0.51$), but neither any significant main effect of the *Augmentation* factor ($F_{(2,30)} = 1.38$ $p > 0.05$, $\eta_p^2 = 0.08$), nor significant *Sea×Augmentation* interaction ($F_{(2,30)} = 0.88$, $p > 0.05$, $\eta_p^2 = 0.05$). Post-hoc tests revealed that the duration of the landing phase was significantly higher in *Rough* than in *Calm sea* (25.12±2.00 vs. 17.72±1.50 sec., $p < 0.05$). Moreover, a paired t-test conducted on the individual average values of the landing phase duration revealed a significant decrease of the landing phase duration between the *Control* and *Addition* modalities in *Rough sea* (27.9 ±8.6 vs. 23.6±6.4 sec., $t_{(15)} = 2.00$, $p < 0.05$). To sum up, the duration of the maneuver, and especially that of the landing phase increased with the sea-state but the duration of the landing phase was lowered by the *Addition* in *Rough sea*.

Therefore, we have further investigated the behavioral origin of this time gain in *Rough sea* between the *Control* and *Addition* modalities.

**3.2.3 Helicopter phase at impact with respect to the ship's heave cycle ($\varphi$).** Helicopter phase at impact with respect to the ship's heave cycle ($\varphi$) was only computed in the *Rough sea* since periodical movements of the ship's deck were observable only in this sea state. The Fig 6 shows that the average inter-individual values of the phase at impact in the pseudo-sinusoidal heave movement of the ship tended to converge toward the moment where the ship started to go upward after a downward movement (phase of landing equal to 204.74±18.84, 204.31 ±16.98 and 206.64±22.01˚ for the *Control*, *Replication* and *Addition* modalities in *Rough sea*,

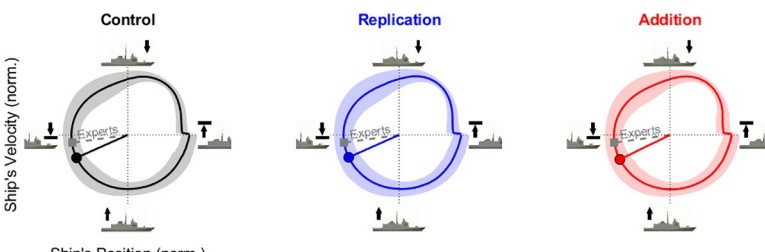

**Fig 6. Inter-individual phase at impact ($\varphi$) on the ship's heave cycle for three *Augmentation* modalities in the *Rough sea*.** The colored line and the shaded area show respectively the inter-individual average and standard deviation of the phase plans computed from the ship's heave movements. The solid radius depicts the inter-individual average value of the phase at impact (in deg.) during the pseudo-sinusoidal vertical movement of the ship across the three augmentation modalities (*Control*, *Replication*, *Addition*, from left to right) of the *Rough sea*. The dotted radius depicts the average value of the phase at impact observed from expert pilots performing astern landings [20].

respectively), independently of Augmentation modality. Hence, despite the fact that we tested novice pilots, the natural inter-individual average value of their phase at impact is close to that achieved by expert pilots. In summary, participants naturally converged toward the optimal phase at impact in the *Rough sea*.

### 3.3 Thrust commands level: Do augmentations allow better control of the engine?

This third section explores whether the augmentations improved the command of the rotor-craft engine. We additionally studied the influence of sea state manipulations. We hypothesized that the improvement of performance with the *Addition* was rooted in an improvement of rotorcraft command. The latter can be due to a more efficient deceleration during the landing phase of the maneuver since the benefits of augmentations were only observed on the duration of that phase.

Fig 7 shows the changes in maximal acceleration with manipulations of the *Sea* and *Augmentations*. A 2-way RM-ANOVA performed on the individual average values of maximum deceleration during the landing phase firstly revealed a significant main effect of *Sea* ($F(1,15) = 23.27$, $p<0.001$, $\eta_p^2 = 0.61$). The amplitude of maximal deceleration was higher (i.e., greater deceleration) in *Rough* than in *Calm* sea ($-3.50\pm0.29$ vs. $-2.33\pm0.20$, $p< 0.05$). Moreover, the ANOVA revealed a significant effect of *Augmentation* ($F(2,30) = 3.98$, $p<0.05$, $\eta_p^2 = 0.21$) but no significant *Sea×Augmentation* interaction ($F(2,30) = 1.02$, $p>0.05$, $\eta_p^2 = 0.06$).

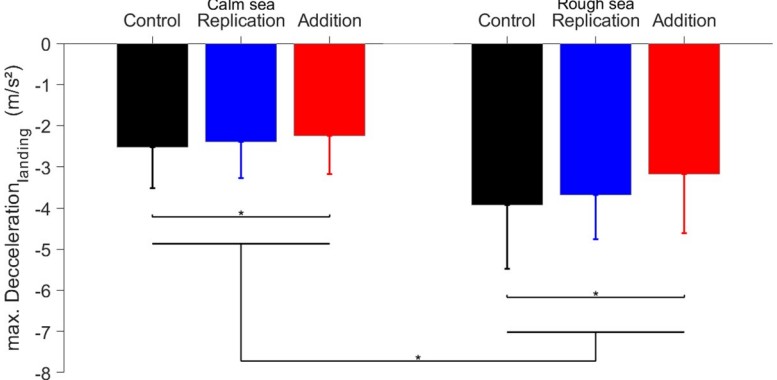

**Fig 7. Inter-individual average values of maximum deceleration during the landing phase.**

Independently of the *Sea* condition, post-hoc tests revealed that the amplitude of maximal deceleration decreased in the *Addition* in comparison with the *Control* and *Replication* modalities (−2.64±0.24 vs. −3.05±1.32, p<0.05). In other words, participants braked less brutally with the *Addition* in comparison to the other modalities. At the same time, the mean deceleration slightly decreased in the *Calm* sea with *Augmentation* manipulations (−0.27±0.02, −0.29±0.02, −0.28±0.02 m/sec.$^2$ for the *Control*, *Replication* and *Addition* modalities respectively), whereas the decrease was more pronounced in the *Rough sea* for the *Replication* and *Addition* modalities (−0.24±0.02, −0.22±0.03 m/sec.$^2$) relative to the *Control* condition (−0.19±0.02 m/sec.$^2$). In summary, the *Addition* allowed both smoother deceleration (i.e., the lower amplitude of maximum deceleration and thus better fine control) and higher mean deceleration values during the landing phase (i.e., higher efficiency of braking).

Moreover, we showed that the improvement of rotorcraft command was also due to a more direct trajectory (i.e., with less backward movements, see Backward Displacement section in S3A Fig in S1 File) and better command of the collective (i.e., with less reversal movement of the joystick, see Actions on collective stick section in S3B Fig in S1 File).

## 3.4 Perceptual-motor strategy level: Do augmentations allow a better coupling of collective stick actions with $\dot{\tau}$?

This last section investigates both whether the design of our augmentations allows a better coupling with $\dot{\tau}$ and how the sea state affects this coupling. Concerning augmentations, previous experimental reports have evidenced that maintaining $\dot{\tau}$ around −0.5 is a smart perceptual-motor strategy to produce smooth and efficient deceleration when visually regulating braking maneuvers. We thus hypothesized that the direct reading of the current value of the $\dot{\tau}$ variable in comparison to the ideal $\dot{\tau}$ value equal to -0.5 in the *Addition* augmentation should help participants to regulate online their deceleration and thus minimize energy at impact when landing on the ship's deck. Consequently, the direct enhancement of the current $\dot{\tau}$ *vs.* ideal $\dot{\tau}$ relationship would help the participants to couple themselves with the ship's deck according to the $\dot{\tau}$ perceptual-motor strategy, and this could explain the participants' performance improvement observed with this augmentation. In addition, we hypothesized that the availability of the current $\dot{\tau}$ value during the full maneuver, without being enhanced in the *Replication* augmentation should, to a lesser extent, help participants regulating online their deceleration and thus explain the lesser improvement of the participants' performance observed with this augmentation. Concerning *Sea*, we hypothesized that the heave of the ship's deck in the *Rough sea* would prevent participants from finely coupling themselves to the deck in accordance to the $\dot{\tau}$−based perceptual-motor strategy.

To investigate how participants adjusted their current $\dot{\tau}$ value as a function of *Sea* and *Augmentation* conditions, we scrutinized the cumulative frequency of collective stick actions as a function of $\dot{\tau}$ at their onset. These curves allow us to investigate the effectiveness of the *Addition* and *Replication* augmentations in the enhancement of a $\dot{\tau}$−based perceptual-motor strategy by analyzing the following three predictions. First, since $\dot{\tau}$−based perceptual-motor strategy is assumed to guide action (i.e., whether to slow down, speed up or keep speed constant), the pull, standby and push collective stick actions would be initiated at specific, relevant current values of $\dot{\tau}$. Second, since the *Addition* augmentation provides a direct reading of the current value of the $\dot{\tau}$ variable in comparison to the $\dot{\tau}$ ideal value (-0.5), it would reinforce the discrimination of the current $\dot{\tau}$ vs. ideal $\dot{\tau}$ relationship and consequently, the slope of the cumulative frequency of collective stick pull, standby and push actions would be steeper than that observed with the *Replication* augmentation, which in turn would be steeper than that observed in the *Control* modality. Thirdly, since a current value of $\dot{\tau}$ equal to -0.5 specifies that

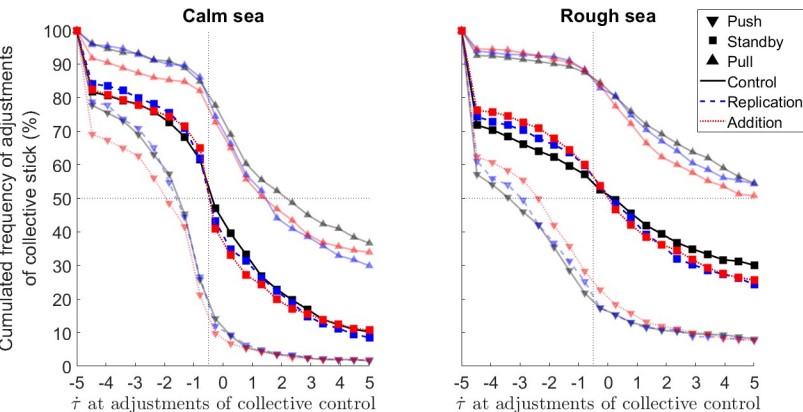

**Fig 8.** Inter-individual average cumulated frequency of pull (i.e., increase velocity, ▲ symbols), standby (i.e., keep velocity constant, * symbols) and push actions on collective stick (i.e., decrease velocity, ▼ symbols) as a function of current $\dot{\tau}$ value at their onset. The solid, dashed, and dotted lines are used to depict *Control*, *Replication*, and *Addition* modalities. The vertical dotted line depicts the ideal $-0.5$ $\dot{\tau}$ value.

the current deceleration would allow a smooth and efficient deceleration, standby collective stick actions would be initiated around that $\dot{\tau}$ value.

Concerning augmentations, Fig 8 shows that for all *Sea* and *Augmentation*s conditions (i.e., when comparing the two panels), the curves for pull, standby and push actions were generally shifting gradually from negative to positive values of $\dot{\tau}$, consistent with our first prediction. Pull actions were used when $\dot{\tau} > -0.5$ to reduce velocity, push actions were performed when the $\dot{\tau} < -0.5$ to increase velocity, and standby actions were produced when $\dot{\tau}$ is equal to $-0.5$ to maintain the velocity. Also, when comparing within a panel the curves of standby actions, the cumulative frequency at 50% were mainly centered around $\dot{\tau}$ equal to $-0.5$ consistent with our second prediction, and the slope of the curve at this moment (which is referred in visual psychophysics as Just Noticeable Difference, JND) increased with the *Replication* and even more with the *Addition* as compared to *Control* modality, consistent with our third prediction. This suggests that the possibility of directly reading of the current value of the $\dot{\tau}$ variable in comparison to the $\dot{\tau} = -0.5$ ideal value in the *Addition* modality allowed participants to regulate the collective stick adjustments more finely.

Concerning the sea state, when comparing pull, standby and push curves between left and right panels, the average slopes of the curves were higher in *Calm sea* than in *Rough sea*. This suggests that actions on the collective stick were more finely tuned as a function of changes in the current $\dot{\tau}$ values in *Calm sea*, than in *Rough sea*.

To quantify these observations, we thus focused on standby actions since they mirrored the strength of the participants' coupling between the actions they made on the collective stick and the $-0.5$ ideal $\dot{\tau}$ value. We extracted the JND (expressed in current $\dot{\tau}$ value) from the individual logistic fits (average $R^2$ values = 0.96, with individual $R^2$ values > 0.71) of the cumulative frequency of standby actions on collective stick. The JND indicates the participants' sensitivity to changes in current $\dot{\tau}$ values for performing the standby action on the collective stick. A 2-way RM-ANOVA performed on the individual average values of JND revealed a significant main effect of the *Sea* (F(1,15) = 22.81, p<0.001, $\eta_p^2$ = 0.60). The JND were significantly stronger in *Calm sea* than in *Rough sea* (-20.84±3.15 vs. -7.86±0.99, p>0.05), suggesting that the participants' sensitivity to changes in $\dot{\tau}$ values were higher in *Calm* than in *Rough sea*. The ANOVA also revealed a significant main effect of the *Augmentation* factor (F(2,30) = 3.47, p<0.05, $\eta_p^2$ = 0.19) but no significant *Sea×Augmentation* interaction (F(2,30) = 1.36, p>0.05, $\eta_p^2$ = 0.08).

Post-hoc tests evidenced that JND significantly sharpened between the *Control* and *Addition* modalities (−11.28±1.81 vs. −18.01±3.30, p<0.05). The sharpened JNDs observed in the *Addition* augmentation in both *Calm sea* and *Rough sea* reveal that displaying the current $\dot{\tau}$ vs. ideal $\dot{\tau}$ relationship on a gauge helped participants to notice any changes in current $\dot{\tau}$ values and in return allowed them to adapt their standby actions on the collective stick as a function of changes to current $\dot{\tau}$ values.

To sum up, standby actions on the collective stick were performed around the $\dot{\tau} = −0.5$ ideal value but were perturbed in *Rough sea* in comparison to *Calm sea*. More importantly, we also noticed that the *Addition* augmentation improved the sensitivity to changes in $\dot{\tau}$ values in comparison to the *Control* modality, when performing standby actions on the collective stick.

## 4. Discussion

In this study, we explored whether the performance of operators completing a ship landing task in a fixed-base helicopter simulator can be improved by feeding their information-movement coupling with additional information. We exposed participants to two different visual augmentations whose designs were either grounded in field analysis and consisted of solving cockpit-induced visual occlusion problems or consisted of strengthening the online regulation of the deceleration by keeping the current $\dot{\tau}$ variable around -0.5. We also manipulated the task's difficulty by exposing participants to *Calm sea* and *Rough sea*. This allowed us to test the effectiveness of two designs of visual augmentation in different weather conditions. The outcome of these manipulations at several levels of analysis (i.e., workload, performance, thrust commands, and information-based landing strategy) are discussed in the following paragraphs.

### 4.1 Ship landing in rough sea

We firstly provided evidence that difficult sea state significantly degraded the participants' performance in landing maneuvers. The substantial increase of Cooper-Harper ratings between *Calm sea* and *Rough sea* suggests an increase of mental workload in the presence of ship heave movements. Participants' performance, expressed as kinematic energy at impact as well as duration of maneuver, was also impaired by *Rough sea*. This performance decrease was linked to a significant decrement of thrust command variables (i.e., occurrence and duration of backward displacements, mean and maximum deceleration) in *Rough sea*, in comparison to *Calm sea*. That is to say, participants were not able to maintain the $\dot{\tau}$ value at −0.5 during their landing maneuver. Such a detrimental influence of sea state in terms of the workload, performance, and behavioral levels of analysis illustrates the legitimacy of research programs trying to find an algorithm to facilitate autonomous landing in a changing environment [56] or to define criteria for ship/helicopter operating limits [57].

Additionally, *Sea* manipulations differently influenced the participants' performance and the use of thrust commands depending on visual augmentation manipulations. Concerning performance, neither approach nor landing phase durations changed significantly with *Replication* and *Addition* as compared to *Control* modality in *Calm sea*. However, the landing phase duration was significantly reduced in *Rough sea* with the *Addition* augmentation as compared to *Control* modality. Concerning thrust commands, the occurrence of backward displacement as well as changes in mean and maximum deceleration also tended to be improved between augmentations when comparing *Calm sea* and *Rough sea*. While we did not evidence any impairment caused by the availability of *Replication* and *Addition* augmentations, enabling augmented reality assistance to landing maneuvers would only be considered in useful situations (e.g., in *Rough sea* and landing phase of the maneuver) as already proposed (see [35] for a

theoretical demonstration of augmentation requirements as a function of Usable-Cue-Environment scale, and [58] for an experimental example of enslavement with visibility). Hence, making visual aids available only when necessary would allow the displays to be used for other purposes without overloading pilots' workload nor occluding field of view.

## 4.2 Design of ecologically grounded augmentations

We secondly evidenced that the *Replication* and *Addition* augmentations improved landing behavior with differing efficiency. The *Replication* augmentation certainly improved landing behavior since it replicated, in the upper section of the participants' field of view, the lowest parts of their outside view (including the deck platform) that are partly or fully masked during the final part of the landing maneuver in *Control* modality. The representation of the ship's deck as a growing circle in the *Replication* augmentation allowed continuing pickup of some perceptual variables of the helicopter-deck system, despite the deck not being visible, and contributed assisting participants to regulate the final part of the landing maneuver. Despite looking stylized due to rendering constraints, the *Replication* augmentation gives rise to a lessening of the cognitive workload relative to the *Control* modality in both *Calm sea* and *Rough sea*. Performance analysis only demonstrated tendencies (and not statistical differences) that the *Replication* augmentation helps to decrease energy at impact. The analyses of thrust commands revealed that the *Replication* augmentation gave rise to statistical improvements in both the maximum and mean accelerations and a significant diminution of backward movements of the helicopter due to a lower number of movement reversals of the collective stick, as compared to the *Control* modality, in the *Rough sea*. These improvements seemed to accompany a finer adjustment of the $\dot{\tau}$ toward the ideal −0.5 value. Taken together, these results indicate the advantage offered by the *Replication* of the deck platform during the final part of the landing maneuver in the *Rough sea*. These improvements, due to the increase in *Replication* augmentation compared to the *Control* modality, thus reflect the sole benefits brought by the cockpit occlusion bypass since the *Replication* display only reproduces the visual scene without facilitating information extraction.

Deceptively, the *Addition* augmentation does not improve the cognitive workload relative to *Control* modalities in either *Calm sea* or *Rough sea*. Verbal reports of participants agreed on preferring the *Replication* to the *Addition* augmentation. Lack of improvement in mental load with the *Addition* augmentation might be rooted in the nature of perceptual processes involved in the regulation of landing. According to the Direct Perception theory [59], perceptual variables like $\dot{\tau}$ are extracted by the user's senses without computation, below the awareness level. These variables are thus used to safely regulate a movement parameter like velocity in the form of laws of control [60]. Therefore, the workload may not have been decreasing with the *Addition* because the $\dot{\tau}$ information was read consciously and merged with the unconscious intrinsic perceptual-motor coupling. It is also possible that the workload did not decrease since we used the Modified Cooper-Harper Handling Qualities Rating Scale that evaluated the physical control of the helicopter and not a scale that shifts the emphasis to how well the augmentations were helping participants [61]. Therefore, it remains to be investigated whether operators perceive our augmentations as facilitating or not their information pickup. The *Addition* augmentation provided a greater improvement to participants' landing behavior over the *Replication*. First, *Addition* allowed the best improvement in performance. Indeed, it did permit a reduction in the duration of the landing phase in comparison with the *Control* condition, in *Rough sea*. The *Addition* augmentation allowed to decrease energy at impact a bit more than the *Replication* did, but not statistically. Analyses of thrust commands revealed that this better performance was probably caused by a decrease in backward displacements, an

increase in mean deceleration, and a reduction of deceleration peaks. These gains cascaded from a decrease in collective stick reversals. Hence, we concluded that the *Addition* augmentation improved the command of the helicopter's engine. These improvements in rotorcraft command are important to consider, bearing in mind that a decrease in the engine load improves safety conditions for pilots by allowing them, with the thus freed engine capacity, a greater margin for action in case of dangerous situations. Finally, we evidenced that this improved behavior was due to an improvement in the sensitivity of changes in $\dot{\tau}$ values when performing standby actions on the collective stick. Therefore, we concluded that the *Addition* provided an optimal improvement of participants' landing behavior probably because it fed the information-movement coupling with the $\dot{\tau}$ variable, thus assisting participants in regulating the velocity gained during the approach phase until touchdown. Information-movement coupling strategies based on maintaining the value of $\dot{\tau}$ around −0.5 have previously been described as efficient when braking in an automobile [31] and also suggested to be useful during helicopter ship landing [35]. Therefore, the *Addition* augmentation may have acted as a smart perceptual instrument, exploiting the power of direct perception [17], to convey in an effective way [16] the current $\dot{\tau}$ vs. ideal −0.5 $\dot{\tau}$ value relationship, assisting participants in performing efficient landings. Benefits were reinforced by the possibility of continuing to pick up the $\dot{\tau}$ value even when the ship deck was not visible, bypassing the cockpit occlusion. Displaying the *Addition* augmentation on a gauge is in the tradition of the first ecologically designed interfaces ([28, 62]) and shows once again the interest of non-projective displays to assist human behavior.

### 4.3 Transfer to real situations, limits and directions for future research

Filling the gap between prototyping visual augmentations in a fixed-base helicopter simulator (i.e., augmented virtuality) and implementing augmented reality interfaces in real situations requires solving a variety of technical problems.

The transfer to real situations firstly requires sensors able to measure in real-time information specifying the current helicopter-ship relationship. The use of artificial vision seems a better option than remote connections from helicopter to ship-board sensors. Indeed, this latter solution would require all ships to be equipped with similar models of sensors, which might be subject to perturbations due to weather conditions and would necessitate encryption for security reasons in a military setting. Artificial vision would moreover allow the helicopter to be independent of the decking surface and would profit from great technological advances made in vision-based approach and landing in aeronautics [63]. A technical choice remains to be made between direct sensing of the relevant information capturing the helicopter-ship relationship ($\dot{\tau}$ in our case) or independent sensing of the different parameters required (cf. parameters used in (1) to compute $\dot{\tau}$).

The transfer to real situations secondly requires display devices to allow pilots to pick up the additional information. Using Head-Up-Display would reduce overload on the pilots' [64] but become useless when pilots already wear see-through displays. In this case, how to merge the augmentation with already existing information remains in question. The way of selecting or automatically displaying the augmentation must also be considered to display augmented information with appropriate timing or in required environmental conditions. Indeed, it may be that different information, or only $\dot{\tau}$, has a different weight on the regulation of the maneuver depending on the landing phase as in [20].

The applicability of these results to real situations is mainly limited by the simplification of the helicopter flight dynamics. First, the degrees of freedom required to command the helicopter were limited on collective stick. This allowed novice participants to fly along a fixed trajectory and prevented their mental workload from being overloaded by rudder or cyclic

command. Second, the collective stick was not operating on a helicopter whose lift capability was bounded by the engine power, the load carried, etc. In real life, a helicopter may thus be understood as being regulated with affordances (see [65] for an illustration of affordance-based models). Therefore, future augmentations should be connected to rotorcraft flight dynamics, to produce changes in trajectories consistent with the action capabilities of the rotorcraft. Thirdly, participants were not managing accuracy since the fixed trajectory always brought them to land on the center of the ship's deck. Finally, note that these results were only gained for an astern landing, with an immobile ship. Other procedures (see [20]), such as oblique, cross-deck and "fore-aft" procedures as well as a vessel navigating on the longitudinal axis should be tested to check the generality of our results. In the future, taking into account a complex model of the ship's heave and not a pseudo sinusoidal model as used in this study to guide the landing with augmented reality would help pilots to better regulate their phase at impact by anticipating future quiescent period of the ship. Further experiments with expert pilots would be required to test these research directions.

## 5. Conclusions

Our approach of design of augmented reality assistance was grounded in EID framework [15]. We showed that feeding the information-movement coupling with the $\dot{\tau}$ variable on a gauge, improved important aspects of landing behavior by significantly reducing the duration of landing maneuver and improving the load on rotorcraft commands. These improvements were probably favored by a finer perception of changes in current $\dot{\tau}$ values that allowed in return finer actions on the collective stick. This is consistent with the direct perception theory [59] as the $\dot{\tau}$ variable specifies the current helicopter-ship relationship and tells pilots whether they are executing a smooth and efficient deceleration leading to a soft landing and how to make corrections if necessary. It moreover confirms, as already suggested, that the $\dot{\tau}$ perceptual-motor strategy can be used during helicopter ship landing [35], as well as in braking tasks while driving [31], hence demonstrating the generalizability across tasks of the perceptual-motor $\dot{\tau}$ strategy. Application of these results to real situations, as well as further investigations of those aspects of landing behavior related to impact that were not improved by the *Addition* augmentation (e.g. energy at impact and relative phase of the touchdown), would however require more realistic rotorcraft dynamics, taking into account the bounded action capabilities of the rotorcraft for guiding landing behavior and thus informing pilots whether the helicopter engine could maintain the $\dot{\tau}$ value around the ideal −0.5 value.

This experiment finally shows that augmented reality is not only a field of application of scientific knowledge but can also constitute a new lever to understand the underlying mechanisms of the visual regulation of behavior, in addition to traditional methodological levers such as visual occlusion and decorrelation paradigms. We argue that tracking the behavioral benefits of visual assistance would act as the "positive symmetry" of measuring the detrimental effects on behavior caused by visual occlusion. Here, the improvement of landing behavior when $\dot{\tau}$ was readable on a gauge not only validates the content and structure of the interface emanating from the work domain analysis but also suggests that this information is not correctly picked up by novices in *Control* modality. This therefore highlights a participants' informational need not provided by their intrinsic capabilities.

## Supporting information

**S1 File.**
(DOCX)

## Acknowledgments

We thank Elena Martin for her work during the preparation of the experiment and the data collection, Christian Schulte and Nawfel Kinani at ONERA for their participation in the Pycs-sHel simulator setup, David Wood (English at your Service, http://www.eays.eu/) for revising the English of the manuscript, Mathieu Thomas for his relevant contributions in the analysis of the landing situation and in the fine understanding of the EID framework, and the five anonymous reviewers for providing thorough and thoughtful critiques of the original manuscript.

## Author Contributions

**Conceptualization:** Antoine H. P. Morice, Thomas Rakotomamonjy, Julien R. Serres, Franck Ruffier.

**Data curation:** Antoine H. P. Morice.

**Formal analysis:** Antoine H. P. Morice.

**Investigation:** Thomas Rakotomamonjy.

**Methodology:** Antoine H. P. Morice, Julien R. Serres, Franck Ruffier.

**Project administration:** Antoine H. P. Morice, Franck Ruffier.

**Resources:** Thomas Rakotomamonjy.

**Software:** Thomas Rakotomamonjy.

**Supervision:** Antoine H. P. Morice, Thomas Rakotomamonjy, Julien R. Serres, Franck Ruffier.

**Validation:** Antoine H. P. Morice.

**Visualization:** Antoine H. P. Morice.

**Writing – original draft:** Antoine H. P. Morice, Thomas Rakotomamonjy, Julien R. Serres, Franck Ruffier.

**Writing – review & editing:** Antoine H. P. Morice, Thomas Rakotomamonjy, Julien R. Serres, Franck Ruffier.

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
