## [Decision Letter · Decision Letter 0]

6 Nov 2020

PONE-D-20-29961

Ecological design of augmentation improves helicopter ship landing maneuvers: an approach in augmented virtuality

PLOS ONE

Dear Dr. MORICE,

Thank you for submitting your manuscript to PLOS ONE. After careful consideration, we feel that it has merit but does not fully meet PLOS ONE’s publication criteria as it currently stands. Therefore, we invite you to submit a revised version of the manuscript that addresses the points raised during the review process.

I have received reviews from four experts. As you will see, there is general agreement about the value of your study. There also are areas that clearly can benefit from revision. Please carefully consider each of the reviews in revising your manuscript. I have confidence that the study will make a useful contribution to the literature.

We look forward to receiving your revised manuscript.

Kind regards,

Thomas A Stoffregen, PhD

Academic Editor

PLOS ONE

Journal Requirements:

2. We note that Figure [1] includes an image of a participant in the study. 

Reviewers' comments:

Reviewer's Responses to Questions

**Comments to the Author**

1. Is the manuscript technically sound, and do the data support the conclusions?

Reviewer #1: Yes

Reviewer #2: Yes

Reviewer #3: Partly

Reviewer #4: Yes

2. Has the statistical analysis been performed appropriately and rigorously? 

Reviewer #1: Yes

Reviewer #2: Yes

Reviewer #3: Yes

Reviewer #4: Yes

3. Have the authors made all data underlying the findings in their manuscript fully available?

Reviewer #1: Yes

Reviewer #2: Yes

Reviewer #3: Yes

Reviewer #4: Yes

4. Is the manuscript presented in an intelligible fashion and written in standard English?

Reviewer #1: Yes

Reviewer #2: Yes

Reviewer #3: No

Reviewer #4: Yes

5. Review Comments to the Author

Reviewer #1: The ecological interface design framework is often referred to as work domain analysis following task analysis to gather information requirements. Identifying user's needs and task analysis are common approaches in human factors, not unique in ecological psychology. To justify the time-to-contact display as an ecological interface design approach, you may describe it based on direct perception and affordance in the introduction.

Fig 4 A & B are hard to read. What is the error bar? Confidence interval or standard deviation? It would be better if the total time and landing duration of maneuver columns could be separate.

Reviewer #2: The authors present an experiment testing a novel display designed to aid helicopter pilots in landing on the deck of a ship heaving in rough seas. A display motivated by Ecological Interface Design (EID) was compared to a display which virtually presented occluded portions of the scene. The ecologically motivated display depicted the current Tau-dot value during the landing approach. Tau-dot is an ecological invariant corresponding to the nature of deceleration relative to an approached surface. Maintenance of an optimal Tau-dot value results in landing with minimal energy at impact. The display depicted in real time the distance of the current Tau-dot value from the optimal Tau-dot value during the landing approach. Performance was improved with both displays, compared to a control condition, with the EID resulting in additional benefits in terms of enhanced aircraft control. This research is both interesting and important. It advances both a theoretical issue and an applied problem. I believe that this line of work should be published, but as described below, I think that the present manuscript requires considerable revision.

Most readers will not be familiar with Ecological Psychology or EID. The Introduction should contain a more thorough description of Ecological Psychology and EID. For example, how is the ecological approach different for more ‘traditional’ approaches to perception? How is the possibility of perception without cognition or representation useful for display design? At the heart of EID is the concept of “mediating direct perception” (Vicente & Rasmussen, 1990, Ecological Psychology, V 2, p 207-249).

The reader is not told about the tested display until the methods section. The display and the task should be described in the introduction. The authors should describe how the invariant is depicted in the display and what information is provided by the display.

Typically, some training period is needed before users are fluent with the use of new (or altered) perceptual information. See the literature on perceptual “calibration.” I think that the efficacy of the display would be better tested if performance was measured after some period of calibration training. In my own work, I have tested displays containing novel information that participants were unable to use before a period of calibration training. That is, during an initial test the display was a complete failure, but then new participants performed well after they were allowed a brief 10-15 minute period of calibration training. Fortunately, the results of the current experiment show that the tested display allows for some degree of performance even without explicit calibration training. However, performance should be improved by calibration training.

Clearly, nobody is going to fly a helicopter (or run a power plant, perform surgery, etc.), without an ample training period. Thus, it seems proper to include training in any evaluation of a novel display. In some past experiments, domain experts with years of experience have been shown to improve their performance after a brief period of calibration training. This is because some perceptional information, including invariants such as Tau, are not always spontaneously used, even by experts. I suspect that the display tested in the present experiment is even more useful than the results presented here indicate. Calibration training may be needed to reveal the display’s full potential.

The task employed in the present experiment involved the participants completing each landing on the simulated ship. Thus, some aspects of calibration training were in fact included in the protocol. The authors should discus the possible role that calibration training played in this experiment, and they should discuss the employment of more thorough calibration training in future investigations. Importantly, the authors should statistically test if there was any improvement in performance seen within the present participants. That is, did they perform better in the later trials compared to the beginning trials?

One thing that is unusual about the present experiment is that novices, with no experience in flying a helicopter, were asked to fly in one of the most challenging scenarios faced by experienced fliers; landing on the deck of a ship in rough seas. An argument could be made for presenting all participants with the calm sea condition first and then the rough seas condition second. This could be one element of calibration training.

The authors employed Ecological Interface Design (EID) by testing a display that provided information regarding a perceptual invariant; Tau-dot. However, EID displays typically present an invariant within the displayed elements. This does not seem to have been done in the present “addition” display. For example, in the groundbreaking paper that did much to establish the field of EID, Vicente & Rasmussen (1990) identified an invariant relationship between the volume, temperature, and energy contained in cooling tank of the type used in power plants and process control facilities. The display they designed was a triangle that changed shape to represent this invariant relationship. It is not clear how the pointer used in the present “addition” display presents to the user the invariant of Tau-dot. The display is akin to a pointer or dial that simply presents the Tau-dot value. It is like if Vicente & Rasmussen simply presented users with an Energy value for a tank, without presenting the users with the underlying invariant relationship between volume, temperature and energy. The invariant relationship is embodied in the triangle. Simply presenting the users with a single value (via a pointer, dial, or numerical display) may be a useful, but it is not EID, and a configural display depicting an invariant relationship may be more effective.

When descending a helicopter onto a surface, or any time a surface is approached, the projection of the surface’s texture elements expand on the retina (or on some hypothetical projection surface). The expansion shows exponential growth when plotted as a function of distance. Thus, if the surface is approached with a constant velocity, the projection of texture elements will grow exponentially. This is the “looming” underling the invariant Tau. One way to decelerate perfectly, so that velocity reaches zero at the same time that distance reached zero, is to decelerate in such a way that the optical expansion of some texture element, such as the “H” inside the circle, expands at a constant rate rather than at an exponential rate. Bees use this invariant in their landings. Have the authors considered using a display that simply shows a circle with an H in it that starts off small and expands as the pilot descends? The pilot could be trained to maintain a constant expansion rate within the display. How useful this would be in rough seas is an open question. This, or some similar configuration that deforms with an invariant relationship, would be better example of EID then the simple pointer.

The idea of using an expanding circle is just an example. It may not be a good solution. The point is that the authors should consider a display that presents an invariant relationship, as Vicente & Rasmussen’s triangle does. I would like to see a more thorough description of what information is contained in the Deck Replication display. What information is contained in the expanding disk inside the ring? To what extent does that information pertain to Tau and/or Tau-dot? It is possible that the augmentations included in the Deck Replication display contain an invariant? Perhaps that display could be modified into a version that contains the invariant, if it does not already contain it, or a version which makes the invariant more salient. Also, as discussed above, calibration training could be used to train the participants to attend to the relevant information contained in that a display.

The display used presently for the “addition” condition, a moving pointer, is the type of display that has been used in the past during calibration training, to train users to calibrate the use of an invariant available in some other display. For example, if you were to test a display consisting of an expanding circle (or some similar EID display akin to Vicente & Rasmussen’s triangle), the pointer could be displayed alongside the circle to indicate to the participant when they are successfully maintaining constant optical expansion (or similar invariant that you have devised). This would help calibrate them to maintaining the constant expansion rate (or other invariant) while braking their descent. In past experiments participants have been trained with the pointer (or some similar feedback) and then tested without the pointer. That is, the pointer is used for training only, and then they are tested for their ability to use the ecological configuration alone.

Reviewer #3: Decision:

Revise and resubmit

Abstract:

1. Overall, I found the abstract to be clear and well-written. The exception being the sentence, “This suggested a need for greater integration of knowledge about psychological understanding in the design of augmented reality systems”. It was not clear to me what the authors were trying to convey.

Introduction:

1. The authors are working on an interesting and important problem. I comment them for their efforts.

2. My main concern about the Introduction is that, other than papers concerning tau-dot, it is unclear how the existing literature informed the present work. In its current form, the Introduction mentions prior work related to landing helicopters on ships, but does not connect it to the authors’ “Replication” or “Addition” augmentations in any meaningful way.

3. On a related note, the Introduction section does not mention publications that appear to be relevant to the present work. A few examples are listed below.

Tušl, M., Rainieri, G., Fraboni, F., De Angelis, M., Depolo, M., Pietrantoni, L., & Pingitore, A. (2020). Helicopter Pilots’ Tasks, Subjective Workload, and the Role of External Visual Cues During Shipboard Landing. Journal of Cognitive Engineering and Decision Making, 14(3), 242-257.

Tritschler, J. K., O'Connor, J. C., Pritchard, J. A., & Wallace, R. (2020). Exploratory Investigation into Rotorcraft Pilot Strategy and Visual Cueing Effects in the Shipboard Environment. Journal of the American Helicopter Society, 65(2), 1-13.

Minotra, D., & Feigh, K. M. (2020). An Analysis of Cognitive Demands in Ship-Based Helicopter-Landing Maneuvers. Journal of the American Helicopter Society.

4. I was also concerned that the authors framed their research in terms of Ecological Interface Design (EID), but do not provide a convincing argument to that effect. EID begins with an analysis of the work domain, followed by the creation of displays that depict that work domain (or aspects of it). The authors argued that EID is a two-step process, beginning with identifying “the operator’s informational needs”, and noted that researchers have conducted task analyses to “investigate pilots’ habits in picking up cues and regulating their maneuvers”. Arguably, work domain analysis is not identifying users’ “informational needs”, and the outputs of those task analyses do not constitute a work domain analysis. Further, the authors argued that the second step in EID is “supplying additional information to the operator”. Arguably, that is an oversimplification of the interface design process in EID, i.e., “supplying additional information to the operator” is not in-and-of-itself “ecological”. Rather, what makes EID displays “ecological” is what they are depicting (e.g., the work domain) and how they are depicting it (e.g., emphasizing system constraints and relations between system components).

To be clear, I am not suggesting that the authors’ research could not, or should not, be framed in terms of EID. I think it might be feasible to do so. For example, I think it is possible to build a case that the work domain includes the need to conduct soft landings on ships. From there, one could argue that conducting soft landings requires deceleration so as to minimize force at impact. From there, one could argue that guiding landings based on tau-dot could accomplish that goal. From there, one could (perhaps) describe how the developed/tested augmentations reflect what makes EID displays “ecological”. For example, one might be able to argue that the “Addition” display depicts the relation between current tau-dot and ideal tau-dot, which provides higher-order information concerning whether pilots are executing a soft landing and, if not, how to correct that. Although, arguably, many non-EID displays are probably designed similarly. My point here is that the authors did not provide a convincing case that their work reflects EID.

5. On a semi-related note, the authors frame their “Addition” display as “improving perceptual attunement to the tau-dot variable”. I do not think that is accurate. Attunement concerns learning to pick up the right information amongst a range of possibilities. Arguably, using the authors’ “Addition” display obviates the need for attunement because the operator can only use the correct information. Further, the authors’ “Addition” display does not depict tau-dot per se. Rather, it depicts current tau-dot vs. ideal tau-dot, which is a higher-order variable that tells operators whether they are executing a soft landing, a too soft landing (i.e., stopping short of the landing point), or a too hard landing. For these reasons, I do not think it is appropriate for the authors to discuss their research/findings in terms of attuning to tau-dot. This comment applies to the rest of the document as well.

6. I had 3 concerns about the writing of the Introduction. I detail those below.

a. The first paragraph of Page 5 makes it seem like the authors are conjecturing that helicopter pilots use tau-dot. Why do that if [18] supports that they do (which is only briefly mentioned)? I think the authors should emphasize [18] more, and de-emphasize their argument that helicopter pilots probably brake in ways that are similar to how automobile drivers brake.

b. On Page 5, the authors wrote “Accessing such additional information can improve the operator’s performance in three ways. First, it can enhance relevant information or highlight regions of the environment unperceived by novice or mentally overloaded operators. Secondly, it can replicate a region of the environment occluded by the cockpit, another vehicle, or by the weather. Thirdly, it can add synthetic information, not available in the real environment.” This sets up the expectation that the authors’ research will address each of these possibilities. As I understand it, the present research concerns the second and third possibilities only. If correct, then the quote above will likely confuse readers.

c. On Page 5, the authors note, in a parenthetical, that their “Replication” display included tau-dot. I thought that warranted a bit more explanation, especially given that later (on Page 9), the authors do not discuss their “Replication” display in terms of tau-dot, and describe their “Addition” display as the “tau-dot Addition” display, which gives the impression that the “Addition” display is the only display to provide tau-dot. On a more general note, I think the “Aims of the present study” section would benefit if the authors would provide more explanation about the nature of their augmentations. Currently, readers only get a brief mention of each, and have to wait several pages before getting more details.

Method:

1. In general, I thought the study was designed and executed well.

2. One exception is that the authors employed a fully within-subjects design when participants could learn during early sessions, and what they learned during those early sessions could carry-over and affect their performance during later sessions. The authors noted that they randomized the order of sessions (with each session constituting one combination of Environment and Augmentation). Randomization does not eliminate carry-over effects though, so it is possible that the authors’ results may, at least partially, reflect carry-over effects.

3. Another exception is that the authors employed Newman-Keuls tests, which, as I understand it, are universally considered to be overly liberal. As such, it is unclear whether results revealed through N-K tests would be supported by alternatives that are less liberal.

Results:

1. I did not have any concerns about how the data were analyzed (beyond the comment above about the use of Newman-Keuls tests).

2. However, I do think the authors could revise Section 3.4 to provide more explanation about what was analyzed, why it was analyzed that way, and what that means. The current version of Section 3.4 does not make that information clear enough, and seems to be written for those who are familiar with methods such as those employed by Yilmaz and Warren (1995).

Discussion and Conclusions:

1. My comments concerning how portions of the Introduction section were framed carry-over to the Discussion section.

2. In Section 4.1, the authors note “Such an influence of environment is well known, but the characterization of its influence at all levels of analysis of participants’ behavior justifies research programs trying to find an algorithm to facilitates autonomous landing in a changing environment (26) or to define criteria for ship/helicopter operating limits (27).” They seem to be arguing that their study adds value because it characterizes the influence of environment “at all levels of analysis”, which could be useful for the purposes stated. If so, and if the authors wish for the present paper to be seen as contributing in that way, then I think the authors should discuss some specifics about how their work could serve those purposes.

3. At the end of Section 4.1, the authors note “Therefore, augmented reality assistance to landing maneuvers must be thought of as removable display that must be enabled only in specific conditions as already proposed (28).” I did not understand why the authors’ results motivate the need for a removable display. None of the results suggest that the presence of the display hindered performance or workload. As such, it is unclear why, other than perhaps pilot preference, it would be necessary to create the augmentation so that it would only be enabled under certain conditions.

4. In Section 4.2, the authors note “The Replication augmentation improved landing behavior …”. I think that is an overstatement. If “landing behavior” means task performance, then the reported results do not support the authors’ conclusion. As I understand it, for each reported analysis, performance with the “Replication” augmentation was equivalent to performance in the “Control” condition, during which participants’ views of the landing pad were occluded.

5. The Conclusions section seems more like content that belongs in the Discussion section than in the Conclusions section. Most of the content of this section concerns a) needing to consider the action capabilities of the aircraft (an idea that has not been mentioned before and does not seem to be directly related to the purpose of the study) and b) that augmentation can be used to understand underlying mechanisms (again, an idea that has not been mentioned before and does not seem to be directly related to the purpose of the study).

6. In the Conclusions section, the authors note “the improvement of landing behavior when ˙ was readable on a gauge suggests that this information is not correctly picked up by novices in Control condition. This therefore highlights a participants’ informational need not provided by their intrinsic capabilities.” As I understand it, in the Control condition, participants’ view of the landing pad were occluded during the landing phase. If so, then the difference between performance in the “Control” and “Addition” conditions could reflect occlusion vs. non-occlusion rather than a shortcoming of participants’ information pick-up capabilities.

Figures:

1. For Figure 2, it would be helpful if all images depicted the same scene, e.g., the ship at the same viewing distance and location within the scene. Also, it would be helpful to see “Calm Sea” and “Rough Sea” versions of each Augmentation (Control, Deck Replication, Addition of tau-dot). Finally, the image quality was quite poor, so it was difficult to discern details related to the augmentations.

Reviewer #4: I think this is a fine piece of work, well worthy of publication. I do have some suggestions that I hope the author will consider.

General comments

1. This is excellent preliminary work for a program of research and development focused on the design of visual aids for helicopter shipboard landing. How does this study fit within a broader R&D framework leading toward incorporating displays of these types into aircraft. I'm not suggesting that entire framework be laid out in painstaking detail in this paper, but some indication of what needs to be done next, where the work needs to go to in the longer term, etc. would provide very helpful context.

2. I was unclear about the nature of the controls that were used in this study. Flying a helicopter is a very difficult skill to acquire, so I'm curious how novices achieved such high levels of control. Clearly, there is a need to conduct a very similar study with pilots who are experts at shipboard landings, with realistic plant dynamics and controls, etc. I couldn't help feeling that the results, though interesting and important, would be even more so coming from an expert population.

3. The display concepts were well described, but I'd like to know a bit about how they were designed. Were pilots involved? This is a key element of ecological interface design, so if there wasn't any pilot involvement in the design process for this work there definitely should be before the work goes much further.

4. I was able to access the data files, in case anyone is keeping track of that.

Specific comments

1."Although augmented reality assistance can be hypothesized to improve

pilots’ performance and the safety of landing maneuvers, especially during difficult weather conditions..."

Although I certainly agree, it would useful to have a brief description of WHY and perhaps HOW augmented reality can be hypothesized to improve performance.

2. "A second difficulty is related to the task’s demands (accuracy of ± 1.5-2 m in position and ±

in azimuth required to land on a 11.5 m wide deck)" - This needs a citation.

3."In this article, we want to study an alternative solution..." - Why is an alternative solution needed? What are the problems with the approaches just described?

4."Without that trick, the reduction of horizontal and vertical FOV (7) can indeed be detrimental for the rotorcraft control." - Are pilots trained to use the door windows? If so, then it's probably not a 'trick', but something a bit more complicated than that. If not, then it seems something a bit closer to a rule of thumb, heuristic, etc. that gets developed over time, passed along by word of mouth?

5."Therefore, several approaches have been proposed to overcome FOV-related problems..." - Just very concisely, how are these approaches working out? Any demonstrated advantages/disadvantages?

6."The objective of designing visual assistance adapted to the users’ needs is intimately linked to aeronautics development and originates in the 1990s (9)." - This sort of work has been going on for a lot longer than that. Stanley Roscoe, for existence, was doing work in this area in the '70s. A lot of work in the '80s as well by quite a few folks. No need to go into it - just FYI.

7."Task analysis has continued, for two decades, to be used to investigate pilots’ habits in picking up cues and regulating their maneuvers" - Closer to 40 years, factoring in Sandy Hart's early work at NASA Ames and others.

8. "This last section investigates whether the design of our augmentations allow a better tuning with..." - Define 'tuning'.

6. PLOS authors have the option to publish the peer review history of their article (what does this mean?). If published, this will include your full peer review and any attached files.

Reviewer #1: No

Reviewer #2: No

Reviewer #3: No

Reviewer #4: **Yes: **Larry Hettinger

---

## [Author Response · Author response to Decision Letter 0]

15 May 2021

Reviewer #1

Reviewer #1: The ecological interface design framework is often referred to as work domain analysis following task analysis to gather information requirements. Identifying user's needs and task analysis are common approaches in human factors, not unique in ecological psychology. To justify the time-to-contact display as an ecological interface design approach, you may describe it based on direct perception and affordance in the introduction.

 We fully rewritten the introduction accordingly with Ecological Interface Design framework. The tested interfaces are now described in line with principles, and methodology promoted by EID, and by the way with direct perception and affordances. The need for time-to-contact-based display in the helicopter deck landing task now results from an original work domain analysis introduced in the revised ms (p. 3-10). 

Fig 4 A & B are hard to read. What is the error bar? Confidence interval or standard deviation? It would be better if the total time and landing duration of maneuver columns could be separate.

 We updated the Figure 4 in the revised ms by depicting the durations of the approach and landing phases rather than the durations of the total maneuver and landing phase. The error bars depict the standard deviation of the individual means. We checked that the figure caption clearly mentions that. The updated figure makes visible the increasing difficulty of the landing phase with the degradation of sea conditions. Indeed, the approach phase lasts the same duration independently of environment and augmentation manipulations whereas the landing phase lasts longer in calm sea than in rough sea and decreases with Addition augmentation. 

Reviewer #2

Reviewer #2: The authors present an experiment testing a novel display designed to aid helicopter pilots in landing on the deck of a ship heaving in rough seas. A display motivated by Ecological Interface Design (EID) was compared to a display which virtually presented occluded portions of the scene. The ecologically motivated display depicted the current Tau-dot value during the landing approach. Tau-dot is an ecological invariant corresponding to the nature of deceleration relative to an approached surface. Maintenance of an optimal Tau-dot value results in landing with minimal energy at impact. The display depicted in real time the distance of the current Tau-dot value from the optimal Tau-dot value during the landing approach. Performance was improved with both displays, compared to a control condition, with the EID resulting in additional benefits in terms of enhanced aircraft control. This research is both interesting and important. It advances both a theoretical issue and an applied problem. I believe that this line of work should be published, but as described below, I think that the present manuscript requires considerable revision.

Most readers will not be familiar with Ecological Psychology or EID. The Introduction should contain a more thorough description of Ecological Psychology and EID. For example, how is the ecological approach different for more ‘traditional’ approaches to perception? How is the possibility of perception without cognition or representation useful for display design? At the heart of EID is the concept of “mediating direct perception” (Vicente & Rasmussen, 1990, Ecological Psychology, V 2, p 207-249).

The first step in designing ecological interfaces is to identify the purposes of the work domain. This is different from cognitive task analysis and physical task analysis, commonly found in user-centered design, in that it searches for information on how the environment works, regardless of the user’s tasks. (An Ecological Approach to Pilot Terrain Awareness). conducting a work domain analysis can be a useful approach to provide the content and structure of an interface that aims at improving pilot terrain awareness.

 The introduction of the revised ms had been deeply rewritten and is now more thorough framed with Ecological Interface Design. As such, we performed an original work domain analysis (see section 1.1.1 “What to display?”, p. 5). 

The reader is not told about the tested display until the methods section. The display and the task should be described in the introduction. The authors should describe how the invariant is depicted in the display and what information is provided by the display.

 We better introduced the candidates perceptual-strategy described in the literature for regulating landing as well as the optical variables on which they are based (see 3rd § of section 1.1.1 “What to display?”, p 7). We now explain how the two “Replication” and “Addition” visual augmentations result from the work domain analysis and the sources of visual information they carry at the end of the introduction of the revised ms (see “How to Dsiplay?” section, p. 8-9). 

Typically, some training period is needed before users are fluent with the use of new (or altered) perceptual information. See the literature on perceptual “calibration.” I think that the efficacy of the display would be better tested if performance was measured after some period of calibration training. According to my own experience , I have already tested displays containing novel information that participants were unable to use before a period of calibration training. That is, during an initial test the display was a complete failure, but then new participants performed well after they were allowed a brief 10-15 minutes period of calibration training. Fortunately, our current results show that the tested display allows for some degrees of performance even without an explicit calibration training session. However, performance should be improved by a calibration training session.

 We noticed in a recent chapter written by Pagano and Day the emphasis made on the role of perceptual calibration [*]. They report that novice participants aiming at using haptic distance-to-break invariant when performing minimal invasive surgery reached a commendable skilled behavior after 15 minutes of calibration training. We call on the reviewer to refer to the response we made to the comment requesting a statistical test about performance improvement during the unfolding of the experiment. We demonstrate that the familiarization phase in our experiment was long enough to calibrate participants.

[*] Pagano CC, Day B. Ecological Interface Design Inspired by “The Meaningful Environment.” In: Wagman JB, Blau JJC, editors. Perception as Information Detection [Internet]. 1st ed. New York, NY : Routledge, 2020. | Series: Resources for ecological psychology: Routledge; 2019 [cited 2020 Dec 18]. p. 37–50. Available from: https://www.taylorfrancis.com/books/9781000054033/chapters/10.4324/9780429316128-4

Clearly, nobody is going to fly a helicopter (or run a power plant, perform surgery, etc.), without an ample training period. Thus, it seems proper to include training in any evaluation of a novel display. In some past experiments, domain experts with years of experience have been shown to improve their performance after a brief period of calibration training. This is because some perceptional information, including invariants such as Tau, are not always spontaneously used, even by experts. I suspect that the display tested in the present experiment is even more useful than the results presented here indicate. Calibration training may be needed to reveal the display’s full potential.

 As stated by the reviewer, even though experts do not spontaneously use perceptual invariants such as Tau. We emphasized this in the introduction of the revised ms (see 3rd § of section 1.1.1 “What to display?”, p 7). Furthermore, external factors due to complication of military operation may obstruct the expert perception-action coupling. Together, these arguments motivate the test of a visual augmentation that replaces the expertise of the operators in the information pickup. We thus designed the Addition augmentation as a visual aid allowing operators that are not able in control situations to pick-up and use to Tau, to perform a direct reading of the relationship between their current value of tau-dot and the ideal tau-dot value and thus decelerate smoothly and thanks to this display. We thus argue that providing a longer training period with the novel display before evaluating it would participate to the attunement of operator to the tau-dot variable (i.e., learning to pick up tau dot in the simulated marine environment) and make the augmentation useless. 

To prevent the complex helicopter motion and commands hide the benefits of the assistance, the control of the helicopter had been simplified at the extreme to allow novices participants to focus on the coupling between the longitudinal (i.e., forward, backward) movements of the helicopter and the visual sources of information emanating in return from the environment. This last point had been enhanced in Introduction (p7) and in Section 2.4 “Task” (p. 12).

The task employed in the present experiment involved the participants completing each landing on the simulated ship. Thus, some aspects of calibration training were in fact included in the protocol. The authors should discuss the possible role that calibration training played in this experiment, and they should discuss the employment of more thorough calibration training in future investigations. Importantly, the authors should statistically test if there was any improvement in performance seen within the present participants. That is, did they perform better in the later trials compared to the beginning trials?

 To convince the reviewer that the practice trials performed during the familiarization phase were sufficient to stabilize the perceptual learning, we analyzed the changes in total duration of the maneuver during the unfolding of the experiment. All analyses are included in the revised supplemental Information (see “Control of perceptual learning during the Familiarization phase” section in Supplementary Information– Fig. S1). Statistical analyses showed that the total duration of the maneuver was significantly longer during the Familiarization phase than during the 1st experimental block and that the total duration of the maneuver does not significantly differ between the 1st and the 6th experimental block (p>0.05). We therefore concluded that the familiarization phase was long enough to allow participants calibrate themselves with the task and Augmentations.

Fig. S1. Inter-individual average total duration of the landing maneuver expressed as a function of the unfolding of trials during the experiment. Trials performed during the familiarization phase are depicted with empty circles, while trials performed during the Experiment phase are depicted with plein circles. The horizontal dotted lines depict the average total duration of the maneuver for the considered block and depict the standard deviation of inter-individual values. Vertical bars depict standard deviation of inter-individual average values of total duration for the considered trial.

One thing that is unusual about the present experiment is that novices, with no experience in flying a helicopter, were asked to fly in one of the most challenging scenarios faced by experienced fliers; landing on the deck of a ship in rough seas. An argument could be made for presenting all participants with the calm sea condition first and then the rough seas condition second. This could be one element of calibration training.

 Indeed, novices cannot control the totality of the degrees of freedom of a real helicopter by using the whole set of commands. Therefore, as explained above and underlined in the revised manuscript (Introduction, page 7 and section 24 "Task", p 12) we asked the participants to control only the speed. Thus, a situation as extreme as landing become similar to a braking task in which novice participants can exploit perceptual-motor control mechanisms used in other contexts. This allowed, as previously demonstrated, to quickly calibrate the participants without the need for additional artifice such as presenting a calm see first. 

The authors employed Ecological Interface Design (EID) by testing a display that provided information regarding a perceptual invariant; Tau-dot. However, EID displays typically present an invariant within the displayed elements. This does not seem to have been done in the present “addition” display. For example, in the groundbreaking paper that did much to establish the field of EID, Vicente & Rasmussen (1990) identified an invariant relationship between the volume, temperature, and energy contained in cooling tank of the type used in power plants and process control facilities. The display they designed was a triangle that changed shape to represent this invariant relationship. It is not clear how the pointer used in the present “addition” display presents to the user the invariant of Tau-dot. The display is akin to a pointer or dial that simply presents the Tau-dot value. It is like if Vicente & Rasmussen simply presented users with an Energy value for a tank, without presenting the users with the underlying invariant relationship between volume, temperature and energy. The invariant relationship is embodied in the triangle. Simply presenting the users with a single value (via a pointer, dial, or numerical display) may be a useful, but it is not EID, and a configural display depicting an invariant relationship may be more effective.

 As stated in the revised manuscript (See for instance section 1.1.2 “How to display?”), the addition display carries the current tau-dot vs ideal-tau-dot relationship. Therefore, this relationship is a higher-order variable that tells pilots’ whether they are executing a smooth and efficient deceleration leading to a soft landing, a too soft landing (i.e., stopping short of the landing point), or a too hard landing (i.e., landing on the deck with a velocity at impact overshooting helicopter structural limitations and spinal column tolerance). It moreover informs participants how to correct their velocity.

When descending a helicopter onto a surface, or any time a surface is approached, the projection of the surface’s texture elements expand on the retina (or on some hypothetical projection surface). The expansion shows exponential growth when plotted as a function of distance. Thus, if the surface is approached with a constant velocity, the projection of texture elements will grow exponentially. This is the “looming” underling the invariant Tau. One way to decelerate perfectly, so that velocity reaches zero at the same time that distance reached zero, is to decelerate in such a way that the optical expansion of some texture element, such as the “H” inside the circle, expands at a constant rate rather than at an exponential rate. Bees use this invariant in their landings. Have the authors considered using a display that simply shows a circle with an H in it that starts off small and expands as the pilot descends? The pilot could be trained to maintain a constant expansion rate within the display. How useful this would be in rough seas is an open question. This, or some similar configuration that deforms with an invariant relationship, would be better example of EID then the simple pointer.

 Concerning alternative strategies to visually control landing, we summed up in the revised manuscript (3rd § of Section 1.1.1 “What to display?”, p.7) the possible candidate optical variables and corresponding strategies to perform the task. The rate of expansion strategy (i.e., maintaining constant the rate of the expansion of texture element) and the supporting experimental evidence in Human and bees were introduced. Moreover, we described how the “Replication” augmentation can enhance this strategy (see answer to the following comments for details about the information carried). 

The idea of using an expanding circle is just an example. It may not be a good solution. The point is that the authors should consider a display that presents an invariant relationship, as Vicente & Rasmussen’s triangle does. I would like to see a more thorough description of what information is contained in the Deck Replication display. What information is contained in the expanding disk inside the ring? To what extent does that information pertain to Tau and/or Tau-dot? It is possible that the augmentations included in the Deck Replication display contain an invariant? Perhaps that display could be modified into a version that contains the invariant, if it does not already contain it, or a version which makes the invariant more salient. Also, as discussed above, calibration training could be used to train the participants to attend to the relevant information contained in that a display.

 As noted above, we claim that the addition display carries a higher order variable that tells pilots whether they are performing a smooth and efficient deceleration leading to a soft landing and how to make corrections if necessary. The reviewer is correct regarding the inaccuracy in the original submission regarding the replication display. This has been corrected in the revised submission. Indeed, the replication display does not only carry the current tau-point vs. ideal-tau-point relationship as the addition display does, the replication display also carries other sources of visual information since it simply reproduces in front of the pilot's field of view the visual scene obstructed by the cockpit. Strictly speaking, the replication display therefore seems more informative than the Addition display. However, it suffers from two drawbacks. First, it does not guide the participants in retrieving the right information. Therefore, participants that do not know what to pickup cannot take any benefits from it. Second, it does not give a direct reading of the invariant like Vincente & Rasmussen triangle or the addition display did. Therefore, participants that do not know how to pickup cannot take any benefits from it. It sole advantage may thus lead in the fact that it allow visual control of landing until the touchdown. 

The display used presently for the “addition” condition, a moving pointer, is the type of display that has been used in the past during calibration training, to train users to calibrate the use of an invariant available in some other display. For example, if you were to test a display consisting of an expanding circle (or some similar EID display akin to Vicente & Rasmussen’s triangle), the pointer could be displayed alongside the circle to indicate to the participant when they are successfully maintaining constant optical expansion (or similar invariant that you have devised). This would help calibrate them to maintaining the constant expansion rate (or other invariant) while braking their descent. In past experiments participants have been trained with the pointer (or some similar feedback) and then tested without the pointer. That is, the pointer is used for training only, and then they are tested for their ability to use the ecological configuration alone.

 We found evidence in the literature that moving pointer like display can be successfully used to help operators to online regulate their locomotion behavior with respect to an ideal value [*]. We used this reference to support the choice we made in the design of the “Addition” augmentation. Moreover, as we hypothesized that tau dot-based strategy would be the most more efficient one, we argue that a display consisting of an expanding circle and assuming an expansion-based strategy would be less powerful. 

[*] Huet M, Camachon C, Fernandez L, Jacobs DM, Montagne G. Self-controlled concurrent feedback and the education of attention towards perceptual invariants. Hum Mov Sci. 2009 Aug;28(4):450–67. 

Reviewer #3

Reviewer #3: Decision:

Revise and resubmit

Abstract:

1. Overall, I found the abstract to be clear and well-written. The exception being the sentence, “This suggested a need for greater integration of knowledge about psychological understanding in the design of augmented reality systems”. It was not clear to me what the authors were trying to convey.

 This sentence had been replaced by a more explicit one : “This underlines the importance for designers of augmented reality systems to collaborate with psychologists to identify the relevant perceptual-motor strategy that must be encouraged before designing an augmentation that will enhance it.” 

Introduction:

1. The authors are working on an interesting and important problem. I comment them for their efforts.

 We thank the reviewer for that comment.

2. My main concern about the Introduction is that, other than papers concerning tau-dot, it is unclear how the existing literature informed the present work. In its current form, the Introduction mentions prior work related to landing helicopters on ships, but does not connect it to the authors’ “Replication” or “Addition” augmentations in any meaningful way.

=> Best to our knowledge no similar distinction between “Replication” and “Addition” of visual content has been presented in the literature so far. The closer paradigm from the “replication” display is called “seeing into the walls” [*] . We tried to better introduce them at the end the introduction of the revised ms.

[*] Azuma RT. A Survey of Augmented Reality. Presence Teleoperators Virtual Environ. 1997 Aug;6(4):355–85.

3. On a related note, the Introduction section does not mention publications that appear to be relevant to the present work. A few examples are listed below.

Tušl, M., Rainieri, G., Fraboni, F., De Angelis, M., Depolo, M., Pietrantoni, L., & Pingitore, A. (2020). Helicopter Pilots’ Tasks, Subjective Workload, and the Role of External Visual Cues During Shipboard Landing. Journal of Cognitive Engineering and Decision Making, 14(3), 242-257.

Tritschler, J. K., O'Connor, J. C., Pritchard, J. A., & Wallace, R. (2020). Exploratory Investigation into Rotorcraft Pilot Strategy and Visual Cueing Effects in the Shipboard Environment. Journal of the American Helicopter Society, 65(2), 1-13.

Minotra, D., & Feigh, K. M. (2020). An Analysis of Cognitive Demands in Ship-Based Helicopter-Landing Maneuvers. Journal of the American Helicopter Society.

 We thank the reviewer for these suggested references. We inserted it in the revised ms.

4. I was also concerned that the authors framed their research in terms of Ecological Interface Design (EID), but do not provide a convincing argument to that effect. EID begins with an analysis of the work domain, followed by the creation of displays that depict that work domain (or aspects of it). The authors argued that EID is a two-step process, beginning with identifying “the operator’s informational needs”, and noted that researchers have conducted task analyses to “investigate pilots’ habits in picking up cues and regulating their maneuvers”. Arguably, work domain analysis is not identifying users’ “informational needs”, and the outputs of those task analyses do not constitute a work domain analysis. Further, the authors argued that the second step in EID is “supplying additional information to the operator”. Arguably, that is an oversimplification of the interface design process in EID, i.e., “supplying additional information to the operator” is not in-and-of-itself “ecological”. Rather, what makes EID displays “ecological” is what they are depicting (e.g., the work domain) and how they are depicting it (e.g., emphasizing system constraints and relations between system components).

 The introduction of the revised ms had been fully rewritten and is now more thorough framed with Ecological Interface Design. As such, we performed an original work domain analysis (see section 1.1.1 “What to display?”, p. 5). The tested interfaces are now described in line with principles, and methodology promoted by EID.

 We thank the reviewer to help us making wording correct. As stated by Vicente and Rasmussen (1992), structuring the problem of designing Ecological Interface calls for a two-step approach (Kim J. Vicente & Rasmussen, 1990; K.J. Vicente & Rasmussen, 1992). This approach makes consensus (Ellerbroek et al., 2009). However, we agree that the two steps were not well described in the original ms. Based on reviewer recommendations, we have corrected what these two steps imply. 

To be clear, I am not suggesting that the authors’ research could not, or should not, be framed in terms of EID. I think it might be feasible to do so. For example, I think it is possible to build a case that the work domain includes the need to conduct soft landings on ships. From there, one could argue that conducting soft landings requires deceleration so as to minimize force at impact. From there, one could argue that guiding landings based on tau-dot could accomplish that goal. From there, one could (perhaps) describe how the developed/tested augmentations reflect what makes EID displays “ecological”. For example, one might be able to argue that the “Addition” display depicts the relation between current tau-dot and ideal tau-dot, which provides higher-order information concerning whether pilots are executing a soft landing and, if not, how to correct that. Although, arguably, many non-EID displays are probably designed similarly. My point here is that the authors did not provide a convincing case that their work reflects EID.

 We thank the reviewer for this guideline we followed. As stated above, the introduction was fully rewritten in line with EID, we introduced an abstraction hierarchy to describe the content and the structure of the work domain and rooted our display on that analyse.

5. On a semi-related note, the authors frame their “Addition” display as “improving perceptual attunement to the tau-dot variable”. I do not think that is accurate. Attunement concerns learning to pick up the right information amongst a range of possibilities. Arguably, using the authors’ “Addition” display obviates the need for attunement because the operator can only use the correct information. Further, the authors’ “Addition” display does not depict tau-dot per se. Rather, it depicts current tau-dot vs. ideal tau-dot, which is a higher-order variable that tells operators whether they are executing a soft landing, a too soft landing (i.e., stopping short of the landing point), or a too hard landing. For these reasons, I do not think it is appropriate for the authors to discuss their research/findings in terms of attuning to tau-dot. This comment applies to the rest of the document as well.

 We agree with the reviewer concerning the abusive property of perceptual attunement, we originally attributed to the addition augmentation. All the involved parts of the original manuscript (Abstract ; section 1.3 “Aim of the present study” ; and Discussion -2nd sentence) were modified as follows: 

 “strengthening the regulation of deceleration by keeping the current τ ˙ variable around the τ ˙=-0.5 ideal value” 

 We also agree that the addition display carries the current tau-dot vs ideal-tau-dot relationship. All the incriminated parts of the manuscript were modified accordingly:

 Example of modification for section 1.3 “Aim of the present study” We hypothesized that pilots’ performance can be improved by the direct reading of the current value of the τ ˙ variable in comparison to the τ ˙=-0.5 ideal value since that relationship is a higher-order variable that tells pilots’ whether they are executing a smooth and efficient deceleration leading to a soft landing, a too soft landing (i.e., stopping short of the landing point), or a too hard landing (i.e., landing on the deck with a velocity at impact overshooting helicopter structural limitations and spinal column tolerance).

6. I had 3 concerns about the writing of the Introduction. I detail those below.

a. The first paragraph of Page 5 makes it seem like the authors are conjecturing that helicopter pilots use tau-dot. Why do that if [18] supports that they do (which is only briefly mentioned)? I think the authors should emphasize [18] more, and de-emphasize their argument that helicopter pilots probably brake in ways that are similar to how automobile drivers brake.

 In the revised ms (last paragraph of the introduction), we de-emphasized the description of the use of tau dot for regulating braking when driving. We also provided a more detailed description of the relevant findings concerning helicopter flight.

b. On Page 5, the authors wrote “Accessing such additional information can improve the operator’s performance in three ways. First, it can enhance relevant information or highlight regions of the environment unperceived by novice or mentally overloaded operators. Secondly, it can replicate a region of the environment occluded by the cockpit, another vehicle, or by the weather. Thirdly, it can add synthetic information, not available in the real environment.” This sets up the expectation that the authors’ research will address each of these possibilities. As I understand it, the present research concerns the second and third possibilities only. If correct, then the quote above will likely confuse readers.

 We rephrased the two last sentences of the revised ms to avoid confusing readers.

c. On Page 5, the authors note, in a parenthetical, that their “Replication” display included tau-dot. I thought that warranted a bit more explanation, especially given that later (on Page 9), the authors do not discuss their “Replication” display in terms of tau-dot, and describe their “Addition” display as the “tau-dot Addition” display, which gives the impression that the “Addition” display is the only display to provide tau-dot. On a more general note, I think the “Aims of the present study” section would benefit if the authors would provide more explanation about the nature of their augmentations. Currently, readers only get a brief mention of each, and have to wait several pages before getting more details

 We described how the “Replication” augmentation can enhance several perceptual-motor strategies (Section 1.1.2 “how to display?” in the revised ms). We also reworked the section 1.2 “aim of the present study” in line with this comment.

Method:

 In general, I thought the study was designed and executed well.

=> thanks for this positive comments.

2. One exception is that the authors employed a fully within-subjects design when participants could learn during early sessions, and what they learned during those early sessions could carry-over and affect their performance during later sessions. The authors noted that they randomized the order of sessions (with each session constituting one combination of Environment and Augmentation). Randomization does not eliminate carry-over effects though, so it is possible that the authors’ results may, at least partially, reflect carry-over effects.

 In our experiment, we ran each combination of Environment and Augmentation conditions into blocks to be able to evaluate the participants’ workload with the Cooper-Harper scale immediate after each block of those combinations. This involved participant rated the task only 6 times during the experiment. If we had not done so, we would not have had a guarantee that the participants' judgements relate to a specific combination of experimental conditions. Given this constraint, since within-subject design are more powerful statistically, we tried to remove the effect of learning by randomizing the order of presentation of blocks. Counter balancing would theoretically eliminate the learning effect, but the participants recruitment was performed as volunteers arise. Since we were not able to predict the final size of the sample, the counter balancement procedure was not adopted. 

 To convince the reviewer that the participants did not learn through the experiment, we analyzed the changes in total duration of the maneuver during the unfolding of the experiment. All analyses are included in the revised supplemental Information (see “Control of perceptual learning during the Familiarization phase” section in Supplementary Information– Fig. S1). Statistical analyses showed that the total duration of the maneuver was significantly longer during the Familiarization phase than during the 1st experimental block and that the total duration of the maneuver does not significantly differ between the 1st and the 6th experimental block (p>0.05). We therefore concluded that randomization eliminated carry-over effects.

Fig. S1. Inter-individual average total duration of the landing maneuver expressed as a function of the unfolding of trials during the experiment. Trials performed during the familiarization phase are depicted with empty circles, while trials performed during the Experiment phase are depicted with plein circles. The horizontal dotted lines depict the average total duration of the maneuver for the considered block and depict the standard deviation of inter-individual values. Vertical bars depict standard deviation of inter-individual average values of total duration for the considered trial.

3. Another exception is that the authors employed Newman-Keuls tests, which, as I understand it, are universally considered to be overly liberal. As such, it is unclear whether results revealed through N-K tests would be supported by alternatives that are less liberal.

 We re-do the whole statistical analysis with Tukey HSD post-hoc tests, which protects the significance tests of all combinations of pairs and reported the results in the revised ms. It appeared that our results do not changed, except for the cooper-harper ratings. In the original ms we reported that “Post-hoc tests revealed that the Cooper-Harper ratings gained with the Replication were significantly lower than those obtained in the Control and Addition condition”. Using the Tukey HSD, we now report in the revised ms that “Post-hoc tests revealed that the Cooper-Harper ratings gained with the Replication were significantly lower than those obtained in the Addition condition”. Therefore, using a more conservative post-hoc do no change conclusions regarding performance, control and perceptual-motor strategy and enhance the positive subjective influence of the Replication condition with respect to the Addition condition. 

Results:

1. I did not have any concerns about how the data were analyzed (beyond the comment above about the use of Newman-Keuls tests).

 Comment processed above.

2. However, I do think the authors could revise Section 3.4 to provide more explanation about what was analyzed, why it was analyzed that way, and what that means. The current version of Section 3.4 does not make that information clear enough, and seems to be written for those who are familiar with methods such as those employed by Yilmaz and Warren (1995).

 We apologize for this inconvenience. We revised the ms by detailing the computations done in the section 2.7.4 of the methods section and by explaining more the meaning of the results in the section 3.4

Discussion and Conclusions:

1. My comments concerning how portions of the Introduction section were framed carry-over to the Discussion section.

 We tried to do our best to keep taking them into account in the revised ms.

2. In Section 4.1, the authors note “Such an influence of environment is well known, but the characterization of its influence at all levels of analysis of participants’ behavior justifies research programs trying to find an algorithm to facilitates autonomous landing in a changing environment (26) or to define criteria for ship/helicopter operating limits (27).” They seem to be arguing that their study adds value because it characterizes the influence of environment “at all levels of analysis”, which could be useful for the purposes stated. If so, and if the authors wish for the present paper to be seen as contributing in that way, then I think the authors should discuss some specifics about how their work could serve those purposes.

 Our study did not focus on the characterization of the influence of environment at several levels of analysis. The sentence had been rephrased. 

3. At the end of Section 4.1, the authors note “Therefore, augmented reality assistance to landing maneuvers must be thought of as removable display that must be enabled only in specific conditions as already proposed (28).” I did not understand why the authors’ results motivate the need for a removable display. None of the results suggest that the presence of the display hindered performance or workload. As such, it is unclear why, other than perhaps pilot preference, it would be necessary to create the augmentation so that it would only be enabled under certain conditions.

 The suggestion we made concerning the need for removable display indeed lies in two observations. First, we reported that the duration of the approach phase was neither influenced by environment nor by augmentation manipulation. Second, we reported differences in participants’ performances and in control of the helicopter engine with the manipulation of the environment. Performances expressed as landing phase duration were not improved by any augmentation (Replication and Addition) in the calm sea environment, while the landing phase duration was significantly reduced in Rough sea with the Addition augmentation as compared to Control condition. Control of the helicopter engine expressed as the occurrence of backward displacement as well as changes in mean and maximum deceleration also tended to be improved with the availability of augmentations when comparing Calm and Rough seas. Taken together, these results suggest that visual aids are not always useful and should only be available in Rough sea environment or during the landing phase. We clarified this in the revised ms (lasts sentences of 4.1 Ship landing in rough sea section) as follow: “While we did not evidence any impairment caused by the availability of Replication and Addition augmentations, enabling augmented reality assistance to landing maneuvers would only be considered in useful situations (e.g., in moderate sea state and landing phase of the maneuver) as already proposed (see Padfield et al., 2003 for a theoretical demonstration of augmentation requirements as a function of Usable-Cue-Environment scale; and 2016 for an experimental example of enslavement with visibility). Hence, thinking visual aids as removable display would allow displaying other augmentations designed for other purposes without overloading pilots workload and occluding field of view.”

4. In Section 4.2, the authors note “The Replication augmentation improved landing behavior …”. I think that is an overstatement. If “landing behavior” means task performance, then the reported results do not support the authors’ conclusion. As I understand it, for each reported analysis, performance with the “Replication” augmentation was equivalent to performance in the “Control” condition, during which participants’ views of the landing pad were occluded.

 The reviewer is right when stating that when considering only the task performance, the replication addition does not statically improve the landing duration. Replication only tends to decrease it. We conveniently corrected this part of our discussion. We nevertheless report statistical improvements of other facets of pilots’ behavior induced by the Replication display (i.e., max and mean deceleration and backward movements). Those improvements of rotorcraft command are important to consider, having in mind that a decrease in the engine load allows pilots benefits from higher actions capabilities if needed, and thus put pilots in increased safety conditions. 

5. The Conclusions section seems more like content that belongs in the Discussion section than in the Conclusions section. Most of the content of this section concerns a) needing to consider the action capabilities of the aircraft (an idea that has not been mentioned before and does not seem to be directly related to the purpose of the study) and b) that augmentation can be used to understand underlying mechanisms (again, an idea that has not been mentioned before and does not seem to be directly related to the purpose of the study).

 We agree with both comments and rephrased the conclusion. In the revised ms, the need for considering action capabilities of the aircraft (i.e. action-scaled affordance models) had been introduced in the introduction (4th level of the abstraction hierarchy, p.6-7), in the discussion of the revised ms (Section 4.6. Transfer to real situations, limits and directions for future research). This idea should now be more in line with the flow of the revised ms. Concerning the understanding of underlying mechanisms, we kept the original idea at its original place, although by reformulating it and connecting it with EID framework. 

6. In the Conclusions section, the authors note “the improvement of landing behavior when ˙ was readable on a gauge suggests that this information is not correctly picked up by novices in Control condition. This therefore highlights a participants’ informational need not provided by their intrinsic capabilities.” As I understand it, in the Control condition, participants’ view of the landing pad were occluded during the landing phase. If so, then the difference between performance in the “Control” and “Addition” conditions could reflect occlusion vs. non-occlusion rather than a shortcoming of participants’ information pick-up capabilities.

 The difference between performances in the control and addition conditions reflect the cumulated influence of occlusion and information feeding. The difference between performance in the “Control” and “Replication” reflect the sole influence of occlusion vs. non-occlusion conditions. We clarified that in the revised discussion (1st § of section 4.2 Design of ecologically grounded Augmentations).

Figures:

1. For Figure 2, it would be helpful if all images depicted the same scene, e.g., the ship at the same viewing distance and location within the scene. Also, it would be helpful to see “Calm Sea” and “Rough Sea” versions of each Augmentation (Control, Deck Replication, Addition of tau-dot). Finally, the image quality was quite poor, so it was difficult to discern details related to the augmentations.

 We are sorry we cannot provide new images due to the complexities of the covid crisis. On the other hand, we have improved the quality of the existing images

Reviewer #4

Reviewer #4: I think this is a fine piece of work, well worthy of publication. I do have some suggestions that I hope the author will consider.

General comments

1. This is excellent preliminary work for a program of research and development focused on the design of visual aids for helicopter shipboard landing. How does this study fit within a broader R&D framework leading toward incorporating displays of these types into aircraft. I'm not suggesting that entire framework be laid out in painstaking detail in this paper, but some indication of what needs to be done next, where the work needs to go to in the longer term, etc. would provide very helpful context.

 Thank you for his comment. The size of the manuscript becoming consequent we could only disseminate some tracks of reflection (e.g., adaptive displays, taking into account of the capacities of actions, application in real situation). These points are discussed in our revised ms (Section 4.6. Transfer to real situations, limits and directions for future research).

2. I was unclear about the nature of the controls that were used in this study. Flying a helicopter is a very difficult skill to acquire, so I'm curious how novices achieved such high levels of control. Clearly, there is a need to conduct a very similar study with pilots who are experts at shipboard landings, with realistic plant dynamics and controls, etc. I couldn't help feeling that the results, though interesting and important, would be even more so coming from an expert population.

 We agree with that comment. In our experiment, the control of the helicopter had been simplified at the extreme to allow novice participants to focus on the coupling between the longitudinal (i.e., forward, backward) movements of the helicopter and the visual sources of information emanating in return from the environment. This difference with real life helicopter flight dynamics had been introduced in the revised ms (4th level of the abstraction hierarchy, p.6-7) and enhanced in Section 2.4 (Task). This simplification prevented facing novices with complex helicopter motion and commands, thus explaining good novices’ performance. 

 We introduced in the discussion of the revised ms (Section 4.6. Transfer to real situations, limits and directions for future research) the need for further experiments with expert pilots.

3. The display concepts were well described, but I'd like to know a bit about how they were designed. Were pilots involved? This is a key element of ecological interface design, so if there wasn't any pilot involvement in the design process for this work there definitely should be before the work goes much further.

 The design of the interfaces was constrained by technical limits in the virtual environment. Only three virtual objects (in addition to the helicopter and frigate) were insertable, enslaving independently on translation and rotation, that could be linked to any virtual object but can not be manipulated during the experiment.

 This experiment was conducted in partnership with ONERA, a French institute that is especially specialized in rotorcraft. It is part of a program in which pilots are interviewed in order to carry out the analysis of the complex field of work

4. I was able to access the data files, in case anyone is keeping track of that.

Specific comments

1."Although augmented reality assistance can be hypothesized to improve pilots’ performance and the safety of landing maneuvers, especially during difficult weather conditions..."

Although I certainly agree, it would be useful to have a brief description of WHY and perhaps HOW augmented reality can be hypothesized to improve performance.

 We modified the abstract as possible to address this comment.

2. "A second difficulty is related to the task’s demands (accuracy of ± 1.5-2 m in position and ±

in azimuth required to land on a 11.5 m wide deck)" - This needs a citation.

 The source [*] has been listed in the revised ms.

[*]Henry B, Mialon B. L’appontage dans la Marine francaise : une approche opérationelle. 2010 Jul p. 63. Report No.: RT 1/166616 DAAP.

3."In this article, we want to study an alternative solution..." - Why is an alternative solution needed? What are the problems with the approaches just described?

 With automatic control, pilots are not inside the loop, limiting their effectiveness in case of complex situations and introducing issues when pilots must regain control. We clarified the corresponding sentences in the revised ms.

4."Without that trick, the reduction of horizontal and vertical FOV (7) can indeed be detrimental for the rotorcraft control." - Are pilots trained to use the door windows? If so, then it's probably not a 'trick', but something a bit more complicated than that. If not, then it seems something a bit closer to a rule of thumb, heuristic, etc. that gets developed over time, passed along by word of mouth?

 We have refined our analysis of the situation, corrected and completed the text with new elements.

5."Therefore, several approaches have been proposed to overcome FOV-related problems..." - Just very concisely, how are these approaches working out? Any demonstrated advantages/disadvantages?

 We briefly stated about pros and cons in the revised ms.

6."The objective of designing visual assistance adapted to the users’ needs is intimately linked to aeronautics development and originates in the 1990s (9)." - This sort of work has been going on for a lot longer than that. Stanley Roscoe, for existence, was doing work in this area in the '70s. A lot of work in the '80s as well by quite a few folks. No need to go into it - just FYI.

 We thank the reviewer for this informative comment.

7."Task analysis has continued, for two decades, to be used to investigate pilots’ habits in picking up cues and regulating their maneuvers" - Closer to 40 years, factoring in Sandy Hart's early work at NASA Ames and others.

 We rephrased the sentence to better take this comment into account.

8. "This last section investigates whether the design of our augmentations allow a better tuning with..." - Define 'tuning'.

 We replaced the ‘tuning’ word by ‘coupling’ as it is better suited in our context.

---

## [Decision Letter · Decision Letter 1]

1 Jun 2021

PONE-D-20-29961R1

Ecological design of augmentation improves helicopter ship landing maneuvers: an approach in augmented virtuality

PLOS ONE

Dear Dr. MORICE,

Thank you for submitting your manuscript to PLOS ONE. After careful consideration, we feel that it has merit but does not fully meet PLOS ONE’s publication criteria as it currently stands. Therefore, we invite you to submit a revised version of the manuscript that addresses the points raised during the review process.

Two of the Reviewers are satisfied with your revisions, but one is not. I agree with Reviewer 3 that substantial improvement is possible, and I feel that it is worthwhile for you to attend to the Reviewer's comments in making final changes to your manuscript.

We look forward to receiving your revised manuscript.

Kind regards,

Thomas A Stoffregen, PhD

Academic Editor

PLOS ONE

Journal Requirements:

Reviewers' comments:

Reviewer's Responses to Questions

**Comments to the Author**

1. If the authors have adequately addressed your comments raised in a previous round of review and you feel that this manuscript is now acceptable for publication, you may indicate that here to bypass the “Comments to the Author” section, enter your conflict of interest statement in the “Confidential to Editor” section, and submit your "Accept" recommendation.

Reviewer #1: All comments have been addressed

Reviewer #2: All comments have been addressed

Reviewer #3: (No Response)

2. Is the manuscript technically sound, and do the data support the conclusions?

Reviewer #1: Yes

Reviewer #2: Yes

Reviewer #3: Partly

3. Has the statistical analysis been performed appropriately and rigorously? 

Reviewer #1: Yes

Reviewer #2: Yes

Reviewer #3: Yes

4. Have the authors made all data underlying the findings in their manuscript fully available?

Reviewer #1: Yes

Reviewer #2: Yes

Reviewer #3: Yes

5. Is the manuscript presented in an intelligible fashion and written in standard English?

Reviewer #1: Yes

Reviewer #2: Yes

Reviewer #3: No

6. Review Comments to the Author

Reviewer #1: Overall, it is well-written. I think it is unnecessary to include the abstraction hierarchy in the introduction if only one aspect is needed. There are different opinions. Either way would work for me.

Example:

Vicente, K. J., Moray, N., Lee, J. D., Hurecon, J. R., Jones, B. G., Brock, R., & Djemil, T. (1996). Evaluation of a Rankine cycle display for nuclear power plant monitoring and diagnosis. Human Factors, 38(3), 506-521.

Reviewer #2: (No Response)

Reviewer #3: The authors extensively revised their manuscript, and, in doing so, addressed many of my concerns. I commend the authors for their efforts.

My remaining concerns are …

1) I think the readability of the manuscript would improve if the authors had a native English-speaker proofread/edit the manuscript.

2) The “What do display?” section discusses many topics, most of which do not have an obvious and direct connection to the displays the authors created/tested. I recommend the authors revise that section to make it more clear how what they described in that section led to the design of their displays.

3) In the “How to display?” Section, it is still not as clear as it could be how the authors’ displays are “ecological”. That is particularly true for the “Replication” display. The section gives the impression that “ecological” mainly applies to the “Addition” display, and may not apply very much (or at all) to the “Replication” display. More generally, I could imagine both the “Replication” and “Addition” displays being designed by someone who does not utilize EID, and the content of this section does not convince me that framing these displays in terms of EID adds value.

4) The paper’s title and Introduction focus heavily on EID. In contrast, the Discussion section surprisingly hardly mentions EID. That gives the impression that EID is not truly the focus, as suggested by the emphasis on EID in the title and Introduction.

5) The authors report that their displays affected certain aspects of landing behavior, and had no effect on other aspects of landing behavior. Despite that variability, they still state, in the Conclusions section (and perhaps elsewhere) that “… feeding the information-movement coupling with the tau variable on a gauge improved landing behavior …”. To me, that statement oversimplifies the situation. The “Addition” display improved certain aspects of landing behavior, but it had no effect on others. Further, some of the aspects of landing behavior that were unaffected presumably should have been affected. For example, the “Addition” displays seems well-suited to improve “Energy at impact”. However, the authors report that only the environment affected “Energy at impact”. When the authors make blanket statements like “The Addition displays improved landing behavior” they are glossing over such nuances. I think the paper would be improved if the authors adequately dealt with such nuances (those related to my examples, but also other reported analyses).

7. PLOS authors have the option to publish the peer review history of their article (what does this mean?). If published, this will include your full peer review and any attached files.

Reviewer #1: No

Reviewer #2: No

Reviewer #3: No

---

## [Author Response · Author response to Decision Letter 1]

18 Jul 2021

Reviewer #1: Overall, it is well-written. I think it is unnecessary to include the abstraction hierarchy in the introduction if only one aspect is needed. There are different opinions. Either way would work for me.

Example:

Vicente, K. J., Moray, N., Lee, J. D., Hurecon, J. R., Jones, B. G., Brock, R., & Djemil, T. (1996). Evaluation of a Rankine cycle display for nuclear power plant monitoring and diagnosis. Human Factors, 38(3), 506-521.

=> Thank you for sharing your feeling. We have summarized the results of the analysis provided by the abstraction hierarchy with a visual sketch. We hope this will help readers better understand the constraints of the work area. 

Reviewer #3: The authors extensively revised their manuscript, and, in doing so, addressed many of my concerns. I commend the authors for their efforts.

My remaining concerns are …

1) I think the readability of the manuscript would improve if the authors had a native English-speaker proofread/edit the manuscript.

=> Thank you for your comment on the language. A professional, native English speaker have edited the reviewed manuscript and figures. We have proofread the manuscript again. 

2) The “What do display?” section discusses many topics, most of which do not have an obvious and direct connection to the displays the authors created/tested. I recommend the authors revise that section to make it more clear how what they described in that section led to the design of their displays.

=> We removed content in this section with care since many topics were asked in previous comments. We clearly explained in the end of the paragraph how the abstraction hierarchy led to the design of our display. We moreover provided an addition sketch up summarizing the results of the analysis provided by the abstraction hierarchy.

3) In the “How to display?” Section, it is still not as clear as it could be how the authors’ displays are “ecological”. That is particularly true for the “Replication” display. The section gives the impression that “ecological” mainly applies to the “Addition” display, and may not apply very much (or at all) to the “Replication” display. More generally, I could imagine both the “Replication” and “Addition” displays being designed by someone who does not utilize EID, and the content of this section does not convince me that framing these displays in terms of EID adds value.

=> Indeed, we are convinced that addition augmentation is more ecological than replication augmentation (and the results attest to this) because not only it bypass the cockpit occlusion, but also it synthesizes tau-dot, an ecological invariant corresponding to the nature of deceleration relative to an approached surface. We reinforced this idea in the description of the “Replication” form.

4) The paper’s title and Introduction focus heavily on EID. In contrast, the Discussion section surprisingly hardly mentions EID. That gives the impression that EID is not truly the focus, as suggested by the emphasis on EID in the title and Introduction.

=> We framed more the existing section 4.2. Design of ecologically grounded Augmentations of the revised discussion on EID.

5) The authors report that their displays affected certain aspects of landing behavior, and had no effect on other aspects of landing behavior. Despite that variability, they still state, in the Conclusions section (and perhaps elsewhere) that “… feeding the information-movement coupling with the tau variable on a gauge improved landing behavior …”. To me, that statement oversimplifies the situation. The “Addition” display improved certain aspects of landing behavior, but it had no effect on others. Further, some of the aspects of landing behavior that were unaffected presumably should have been affected. For example, the “Addition” displays seems well-suited to improve “Energy at impact”. However, the authors report that only the environment affected “Energy at impact”. When the authors make blanket statements like “The Addition displays improved landing behavior” they are glossing over such nuances. I think the paper would be improved if the authors adequately dealt with such nuances (those related to my examples, but also other reported analyses)

=> The reviewer has fully understood the all the results reported. We followed the reviewer’s recommendations and removed the overstatements. As stated in the discussion of the revised manuscript, the Addition « improved important aspects of landing behavior by significantly reducing the duration of landing maneuver and improving the load on rotorcraft commands. These improvements were probably favored by a finer perception of changes in current τ ˙ values that allowed in return finer actions on the collective stick». These improvements are important to consider. Not only the first level of the abstraction hierarchy reveals that minimizing flight time allows to economize fuel (section 1.1.1 What to display ?, p. 7) but also improvements in rotorcraft command allows a decrease in the engine load that improves safety conditions for pilots by allowing them, with the thus freed engine capacity, a greater margin for action in case of dangerous situations (last § of section 4.2. Design of ecologically grounded Augmentations, p 39). 

We also stated in the conclusion of the revised ms that significant improvements were not observed on « aspects of landing behavior related to impact (i.e., relative phase of the touchdown, energy at impact) were not significantly improved by the Addition augmentation”. The reviewer can admit that in the Rough Sea modality, where the minimization of impact energy is important to control because of the heave motion of the ship, the variables energy at impact and relative phase of the touchdown change, albeit very slightly, but still in the direction expected by the benefit brought by the augmentation. We attribute these results to the simplified helicopter model.

---

## [Editor Report · Decision Letter 2]

26 Jul 2021

Ecological design of augmentation improves helicopter ship landing maneuvers: an approach in augmented virtuality

PONE-D-20-29961R2

Dear Dr. MORICE,

We’re pleased to inform you that your manuscript has been judged scientifically suitable for publication and will be formally accepted for publication once it meets all outstanding technical requirements.

Kind regards,

Thomas A Stoffregen, PhD

Academic Editor

PLOS ONE
---

## [Editor Report · Acceptance letter]

2 Aug 2021

PONE-D-20-29961R2 

Ecological design of augmentation improves helicopter ship landing maneuvers: an approach in augmented virtuality 

Dear Dr. MORICE:

I'm pleased to inform you that your manuscript has been deemed suitable for publication in PLOS ONE. Congratulations! Your manuscript is now with our production department. 

Kind regards, 

on behalf of

Dr. Thomas A Stoffregen 

Academic Editor

PLOS ONE